# Exploring perceptual straightness in learned visual representations

**Anne Harrington**[1,2]    **Vasha DuTell**[1,2]    **Ayush Tewari**[1]    **Mark Hamilton**[1]
**Simon Stent**[3]    **Ruth Rosenholtz**[1,2]    **William T. Freeman**[1]
[1]MIT CSAIL    [2]MIT Brain and Cognitive Sciences    [3] Woven Planet
{annekh,vasha}@mit.edu

## Abstract

Humans have been shown to use a "straightened" encoding to represent the natural visual world as it evolves in time (Hénaff et al. 2019). In the context of discrete video sequences, "straightened" means that changes between frames follow a more linear path in representation space at progressively deeper levels of processing. While deep convolutional networks are often proposed as models of human visual processing, many do not straighten natural videos. In this paper, we explore the relationship between network architecture, differing types of robustness, biologically-inspired filtering mechanisms, and representational straightness in response to time-varying input; we identify strengths and limitations of straightness as a useful way of evaluating neural network representations. We find that (1) adversarial training leads to straighter representations in both CNN and transformer-based architectures but (2) this effect is task-dependent, not generalizing to tasks such as segmentation and frame-prediction, where straight representations are not favorable for predictions; and nor to other types of robustness. In addition, (3) straighter representations impart temporal stability to class predictions, even for out-of-distribution data. Finally, (4) biologically-inspired elements increase straightness in the early stages of a network, but do not guarantee increased straightness in downstream layers of CNNs. We show that straightness is an easily computed measure of representational robustness and stability, as well as a hallmark of human representations with benefits for computer vision models.

## 1 Introduction

Visual input from the natural world evolves over time, and this change over time can be thought of as a trajectory in some representation space. For humans, this trajectory has a different representation at varying stages of processing, from input at the retina to brain regions such as V1 and finally to perception (Fig 1). We can ask whether there are advantages to representing the natural evolution of the visual input over time with a straighter, less curved, trajectory. If so, one might expect that human vision does this. (Hénaff et al., 2019) demonstrated that trajectories are straighter in human perceptual space than in pixel space, and suggest that a straighter representation may be useful for visual tasks that require extrapolation, such as predicting the future visual state of the world.

Learning a useful visual representation is one of the major goals of computer vision. Properties like temporal stability, robustness to transformations, and generalization – all of which characterize human vision – are often desirable in computer vision representations. Yet, many existing computer vision models still fail to capture aspects of human vision, despite achieving high accuracy on visual tasks like recognition (Feather et al., 2019; Hénaff et al., 2019). In (Hénaff et al., 2019) it was found that, while biologically-inspired V1-like transformations yield straighter representations compared to the input domain, popular computer vision models such as the original ImageNet-trained AlexNet (Krizhevsky et al., 2017) do not.

In an effort to achieve favorable human-vision properties for computer vision models, there has been much work dedicated to incorporating various aspects of human vision into deep neural networks. These include modifying network architectures to mimic aspects of the human visual system Huang & Rao (2011), for example incorporating filter banks similar to the receptive field properties of visual

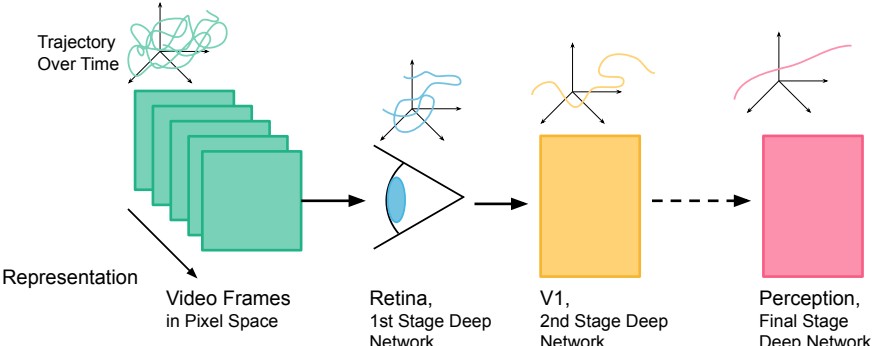

Figure 1: Schematic illustration of the representation of a discrete video sequence becoming progressively straighter as information is processed through a visual processing pipeline, starting from the highly nonlinear trajectory of typical video frames in pixel space.

neurons Dapello et al. (2020), as well as enforcing activation properties similar to those seen in visual cortex such as sparsity Wen et al. (2016). Another promising avenue has been in modifying training to include adversarial examples. By directly targeting areas of vulnerability, adversarially robust networks show increased representational robustness, more closely aligning their performance with that of their human counterparts (Engstrom et al., 2019b).

To understand if attempts to improve neural network models lead to representational straightness, we explore how well models with different architectures, training schemes, and tasks straighten temporal sequences. We evaluate a variety of network architectures, both biologically and non-biologically inspired, for representational straightness across layers; we then ask whether training for adversarial robustness in both CNN and transformer-based architectures may lead to the straighter representations generated by human vision. Because DNNs learn an early representation that differs from what is known about human vision, we also ask if hard-coding that early representation might lead to a trained network with more straightening downstream. We find that straightness is a useful tool that can give intuitions about what allows models to adopt representations that mirror the advantageous qualities of human vision, such as stability.

## 2  PREVIOUS WORK

Deep neural networks have been proposed as models of human visual processing, owing to their ability to predict neural response patterns (Yamins & DiCarlo, 2016; Rajalingham et al., 2015; Kell & McDermott, 2019). As such, there has been much effort to improve the alignment of deep networks to human vision by incorporating known aspects of the human visual system. Some of these include: simulating the multi-scale V1 receptive fields of early vision (Dapello et al., 2020), adding foveation using a texture-like representation in the periphery at a CNN's input stage (Deza & Konkle, 2020), and incorporating activation properties of visual neurons such as sparsity Olshausen & Field (1997); Wen et al. (2016). Predictive coding, often attributed to biological networks Huang & Rao (2011), has been incorporated into deep networks trained to perform tasks such as video frame prediction Lotter et al. (2016), using layers that propagate error-signals.

The desire to evaluate the effectiveness of these techniques at creating models of the human visual system, motivated the creation of measures like BrainScore Schrimpf et al. (2020) that compare models to humans using neural and behavior data. In addition, a number of perceptual experiments have been used to compare human and model representations (Berardino et al., 2017; Feather et al., 2019; Harrington & Deza, 2022). However, perceptual approaches in particular can require lengthy stimuli synthesis procedures and the use of human participants to probe each model's representation. The straightness/curvature measure, however, is a quick and computationally powerful quantitative measure of how well a model aligns with properties of human visual representations Hénaff et al. (2019), particularly in terms of temporal stability.

One important area in understanding how humans and DNNs differ lies in their response to adversarial examples (Elsayed et al., 2018; Ilyas et al., 2019; Feather et al., 2022; Dapello et al., 2021). Adversarial examples, which modify images with changes that are imperceptible to humans, can

cause a network to misclassify an image (Goodfellow et al., 2014; Szegedy et al., 2013). Adversarial training (Madry et al., 2017) improves the misclassfication problem and has been suggested to help networks learn visual representations that are more perceptually aligned with humans Engstrom et al. (2019b); Ilyas et al. (2019). If adversarial training leads to models that are more aligned with human perception, perhaps they also learn straight representations like humans do. Adversarial training schemes, however, are not biologically plausible, and recent work has identified mechanisms that are better supported by vision science (Dapello et al., 2020; Guo et al., 2022; Dapello et al., 2021).

In the context of representation learning in computer vision, adversarially robust models have also been shown to do better at transfer learning than their non-adversarially robust counterparts (Davchev et al., 2019), and adversarially robust features can be used directly for tasks like image generation and in-painting (Santurkar et al., 2019). Adversarial training schemes have also been developed for tasks other than classification like semantic segmentation (Xu et al., 2021). In this paper, we build on work around adversarial robustness by evaluating if increasing this robustness leads to straightened representations like those found in human spatiotemporal processing.

There have been various efforts to incorporate constraints that reduce curvature and increase straightness in temporal representation learning. Slow feature analysis (Berkes & Wiskott, 2005) encourages a learned representation to change slowly over time, but does not encourage straightness explicitly. Another early attempt to add a straightness constraint to representation learning (Goroshin et al., 2015) focused on video frame prediction for unsupervised learning. We follow in the same vein of this work, evaluating straightness as a metric for encouraging a robust, stable, representation when learning to solve tasks.

## 3 METHODS: MEASURING STRAIGHTNESS AS LOW CURVATURE

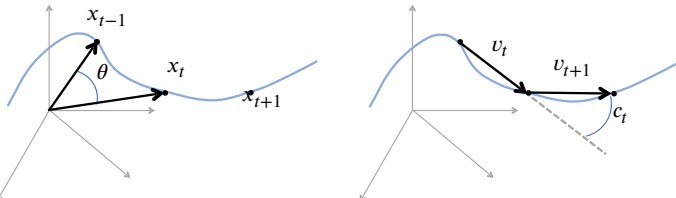

Figure 2: Illustration of how the curvature measure is distinct from cosine similarity. (Left) Three points are sampled along a trajectory in time $(x_{t-1}, x_t, x_{t+1})$. The angle $\theta$ between neighboring $x$ samples is their cosine similarity. (Right) Curvature $c_t$ is the angle between $v_1$ and $v_{t+1}$, where $v_t$ is the difference between $x_t$ and $x_{t-1}$.

Representational straightness can be evaluated as a reduction in curvature. For a temporal sequence, such as a video, curvature is defined as the angle between the vectors representing the *difference* between consecutive frames. Let $\mathbf{x}$ refer to a representation of a video of length $T$, with $x_t$ being a representation of one frame of a video at time step $t$. The representation may be at any stage of the processing pipeline, from a vector of raw input pixels from the video frame, to the activations of a network's hidden layer. Then, $v_t$ represents the difference between successive frames. We can find the curvature at time $t$ by finding the angle between successive $\hat{v}_t$, which we call $c_t$ (equation 1).The global curvature of a video sequence is then simply the mean angle over all time steps. Note there are $T-1$ time steps due to curvature being evaluated on frame differences (equation 2).

$$v_t = x_t - x_{t-1}, \quad \hat{v}_t = \frac{v_t}{\|v_t\|} \quad c_t = \arccos\left(\hat{v}_t \cdot \hat{v}_{t+1}\right) \tag{1}$$

$$\text{Global curvature} := \frac{1}{T-1} \sum_{t=0}^{T-1} c_t \tag{2}$$

This is the formulation proposed by (Hénaff et al., 2019). One can compute this global curvature for any representation of a video sequence over time, either on the vector of pixels (likely not very straight), or one can apply it to a representation of that video, e.g. at any layer of a neural network model. In this paper, we often report change in curvature with respect to the pixel-value input to show if a model's representation is more straight at later stages of processing. Change in curvature is the representational curvature minus the pixel curvature.

Note that curvature is distinct from simple cosine similarity in that curvature is calculated on frame differences ($v_t$), whereas cosine similarity depends on the angle between the frame vectors themselves ($x_t$). Curvature can be thought of as a first-order variant of cosine similarity (see Figure 2 and Sec A.1). For a complete description of how curvature was determined for each model, see Sec A.2.2. We also explore the effect of principle components analysis on curvature in Sec A.2.4 and provide a note on compute requirements in Section A.2.3.

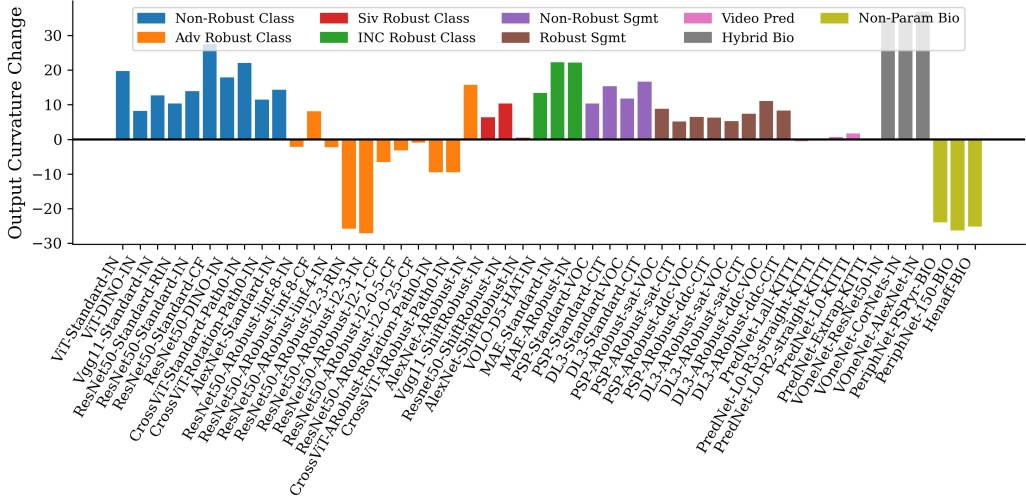

Figure 3: **Output Representation layer curvature for a variety of network architectures and training schemes.** The x-axis ticks describe different models and are formatted as: [architecture]_[training scheme]_[training dataset]. The datasets are IN:ImageNet (Deng et al., 2009), RIN:RestrictedImageNet (Santurkar et al., 2019),CF:CIFAR10 (Krizhevsky et al., 2014), VOC:PASCAL VOC 2012 (Everingham et al., 2010), CIT:Cityscapes (Cordts et al., 2016), KITTI:KITTI (Geiger et al., 2012). Exact model details can be found in the supplemental data. The y-axis shows curvature of each model at the output representation layer relative to the curvature of the input video. For image classification models the output representation is the penultimate layer.

## 4 CURVATURE AND ROBUSTNESS IN CONVOLUTIONAL OBJECT RECOGNITION MODELS

We measured curvature for a variety of models (Sec A.2.1) to investigate the relationship between model type and output curvature. As shown in Figure 3, we find non-adversarially trained image recognition models have the highest output curvature. All adversarially trained models have lower curvature than their non-adversarially trained counterparts (Sec A.4), as well as overall, with the majority reducing output curvature *below* that of the input pixels. Self-supervised DINO (Caron et al., 2021) models have similar output curvature values to their supervised counterparts – despite DINO models having been shown to have more semantically meaningful feature correspondences.

### 4.1 ADVERSARIAL ATTACK TYPE, STRENGTH, AND CURVATURE

Given the increased straightness seen in adversarially robust image recognition / object detection models, we investigated the relationship between the type and strength of adversarial attack and the resulting curvature of the model's output. To evaluate the effect of these attacks on curvature, we compare a set of ResNet50 networks (He et al., 2016), trained on CIFAR-10 (Krizhevsky et al., 2014), ImageNet (Deng et al., 2009), and Restricted ImageNet (a subset of ImageNet (Engstrom et al., 2019b; Ilyas et al., 2019)), with and without adversarial training and measure the output curvature (Fig 5, Left). The adversarially trained models are trained using projected gradient descent with $l_2$ or $l_\infty$ norms at different perturbation levels (Madry et al., 2017; Engstrom et al., 2019a). We use ResNet50 adversarially trained with projected gradient descent because that model type has been shown to be more aligned with human perception (Engstrom et al., 2019b; Feather et al., 2022;

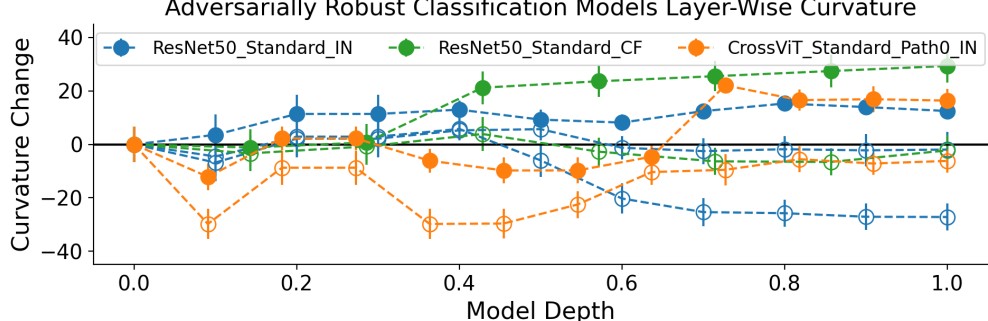

Figure 4: **Deep network models vary in curvature for each layer over model depth.** Curvature is shown for a variety of model. Filled circles indicate non-adversarially trained models, and open circles indicate adversarially-trained robust models. All adversarial models show output curvature below zero, indicating output straightening. All non-adversarial models have output curvature above zero, indicating output representation is *less* straight than the input pixels; All adversarial models shown have output curvature below zero, indicating output representation is *more* straight than input.

Harrington & Deza, 2022). During projected gradient descent, adversarial examples are created to train the model to correctly classify them. The adversarial example is created by adding noise to an image. The amount of noise is bounded by a norm (usually $l_2$ or $l_\infty$ set to a level $\epsilon$). We include some models trained on CIFAR-10 to understand how perturbation level $\epsilon$ interacts with straightness (CIFAR-10 models are faster to train, so more models are available).

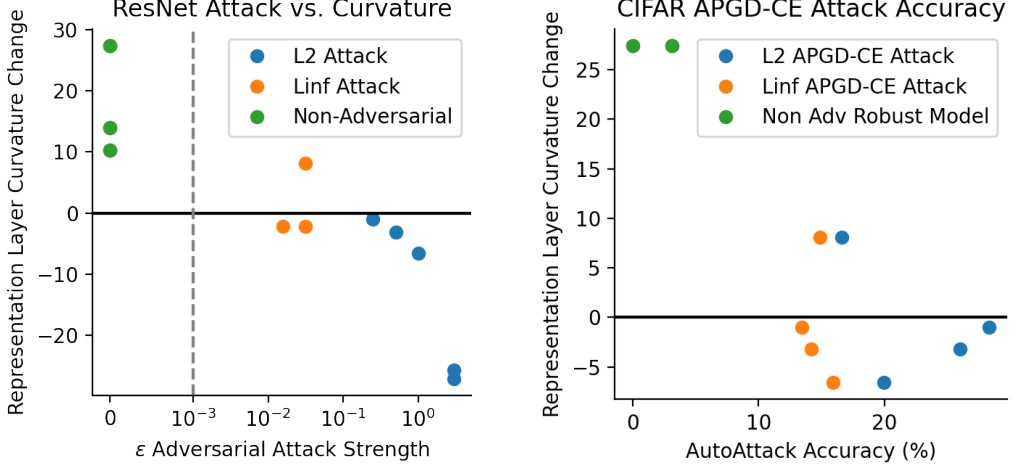

Figure 5: **Left**: White-box adversarial attacks reduce curvature in the output layer of ResNet50 models. Increased attack strength ($\epsilon$) decreases curvature, with $l_2$ attacks most reducing curvature in the resulting model. Data on symlog x-scale to show $\epsilon = 0$: data to left of line on linear scale to show $\epsilon = 0$, and right of line on log scale. **Right**: For a variety of robustly trained models, Improved AutoAttack accuracy for APGD-CE is predicted by larger decreases in curvature from the pixel layer for adversarially robust CIFAR-trained ResNet50 models. We plot CIFAR-trained models trained with $l_2$ norm, $\epsilon = 0, 0.25, 0.5, 1.0$.

We find that, as predicted, output curvature is highest for non-adversarially trained networks. $l_\infty$ attacks decrease output curvature, with larger values of $\epsilon$ leading to decreased curvature. $l_2$-attacked networks have the straightest output curvatures, however strength of attack does not greatly affect the output curvature. The $l_2$ norm models may achieve straighter representations than the $l_\infty$ models because they allow for greater $\epsilon$ in training. Among $l_2$ models, higher $\epsilon$ values may need to be tested to see a difference in curvature among $l_2$ norm trained models. Overall, we show that adversarial robustness to larger perturbations leads to straighter representations.

While curvature corresponds to the strength of adversarial attack a network has been trained with, increased adversarial attack strength does not often translate to improved test set accuracy (Tsipras et al., 2018) (Sec A.5). In order to evaluate the predictive power of curvature for adversarial accuracy, we used the AutoAttack toolbox (Croce & Hein, 2020) to perform APGD-CE attacks on the CIFAR-trained subset of the Robust ResNet50 models (Fig 5, Right). When comparing the Accuracy scores for these models with their curvature, we find that for both $l_2$ and $l_\infty$ attacks, more curvature reduction in the representation layer is correlated with higher attack accuracy.

We also explore how curvature relates to other types of robustness by evaluating leaders on the ImageNet-c benchmark (Hendrycks & Dietterich, 2019) and anti-aliasing CNNs (Zhang, 2019). We find that none of these other robust models straightened natural videos like the adversarially robust models (See Fig 24 for corruption robust models: MAE, MAE-DAT, YOLO-d5-HAT; See Fig 25 for a comparison of ResNet50 standard, shift-invariant robust, and adversarially robust; See Fig 27 for anti-aliasing ResNet50, VGG, AlexNet). Even though models trained for other types of robustness behave more like humans, the curvature measure reveals that those types of robustness are not sufficient to create the temporal representational stability observed in humans (Hénaff et al., 2019). Adversarial robustness may stand-out as producing straightness because it corresponds to more human-meaningful representations as shown with feature inversion (Engstrom et al., 2019b).

## 4.2 CURVATURE AND REPRESENTATIONAL STABILITY

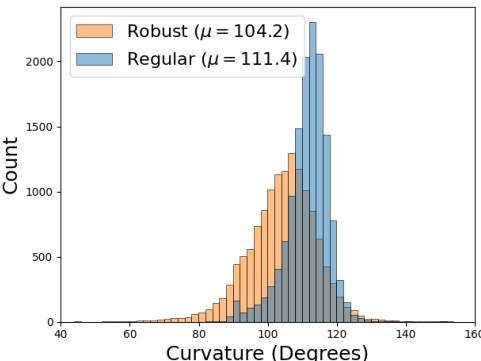 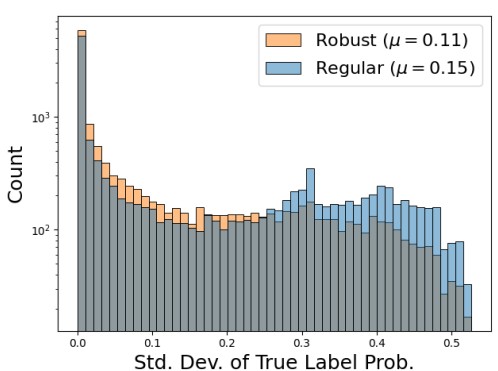

Figure 6: **Left**: Curvature distribution for Adversarially Robust (Robust) and Non-Robust (Regular) ResNet when tested on VGGSound dataset. Robust model shows lower curvature over all videos. **Right**: Standard Deviation of true label prediction for same networks. Robust model shows overall lower variability in probability for the true label.

One potential benefit of lower curvature is a stable representation over time. Such a stable representation should show increased stability of class predictions over the frames of a video. To test this, we tested an adversarially robust (ResNet50_Robust_l2_3_IN) and non-robust (ResNet50_Standard_IN) model on VGGSound (Chen et al., 2020). We measured curvature in the final layer for each video, and the standard deviation of the predicted probability of the ground truth labels for the video (Fig 6). The adversarially robust ResNet50 produces an overall lower curvature than the standard-trained model over the VGGSound videos. Furthermore, it has an overall lower standard deviation in predicted probability for the true labels. This indicates that adversarially robust models, which have lower curvature, also show more stable predictions over time, despite the tested videos being out-of-distribution from the training set of images. Thus, curvature can used as a computationally cheap proxy for the temporal stability of network representations and predictions.

## 5 CURVATURE IN VISUAL TRANSFORMER MODELS FOR OBJECT RECOGNITION

Vision transformers (ViT) have very different architectures than CNNs. They replace convolutional elements with self-attention on visual tokens and achieve state of the art performance on a variety of visual tasks. In addition, these networks are more adversarially robust than standard-trained CNNs, which is attributed to their higher-level and more generalizable features (Shao et al., 2021). Despite this, we find that standard-trained ViT (ViT Base, patch 16) does not reduce curvature on

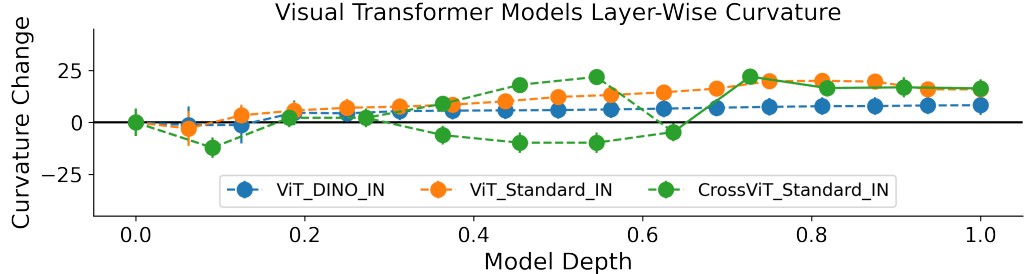

Figure 7: **Visual Transformer models vary in curvature over model blocks.** All ViT models (standard and DINO, base-patch-16) show output curvature higher than the pixel input. For DINO and standard ViT models, this increase is consistent throughout the layers. For CrossViT (18 dagger 408) the representation forks in the middle stage. One path increases in curvature, while the other decreases below input, but after recombination output is curved above pixel input.

the natural video sequences tested (Fig 3). Self-supervised training of ViT (base, patch 16) using the DINO method (Caron et al., 2021) also results in a more curved representation than the input space. However, this DINO trained ViT has a more straight representation than the baseline ViT. This suggests that self-supervised training may be better than supervised for straight representations.

No transformer model except for CrossViT daggar (Chen et al., 2021) reduced curvature in any layer compared to the pixel input (Fig 7). The structure of CrossViT models is unique in that they split into multiple paths, and the reduction in curvature we found interestingly only happens in one pathway of the model and does not persist after the two paths recombine. A possible explanation for this is that the CrossViT is able to straighten in the part of the model that splits because that pathway utilizes a multi-scale representation, a known aspect of human vision representations such as V1 which we show in Sec 7 to induce straightening.

## 6 CURVATURE AND ROBUSTNESS IN IMAGE SEGMENTATION AND FRAME PREDICTION MODELS

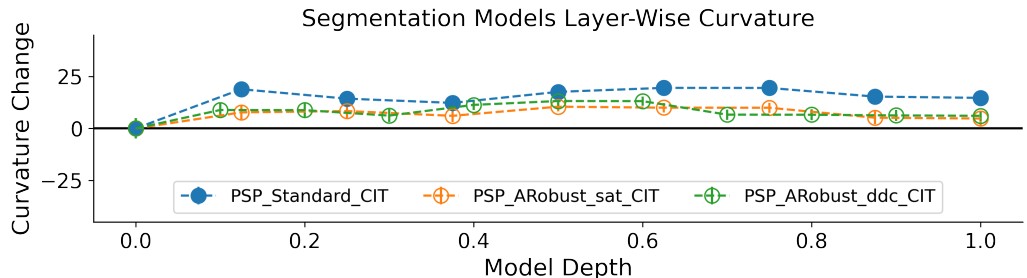

Figure 8: **Segmentation models vary in curvature for each layer over model depth.** Curvature is shown for various segmentation model types. Filled circles indicate non-adversarially trained models, and open circles indicate adversarially-trained robust models.

Output straightness over time should be a favorable, stabilizing property for object detection models, where objects often stay continuously in-frame over many frames of a video. For image segmentation models on the other hand, predictions lie close to low-level pixel space and change on a frame-by-frame basis. Therefore, for the task of predicting the dynamically changing segmentation map of an input movie, straightness of the output representation would *not* be desirable (assuming the input movie contains motion). To test this prediction, we measured output curvature for a family of 12 image segmentation models from (Xu et al., 2021), consisting of two different architectures, trained on two datasets, with a non-adversarial and two different adversarial training schemes.

Unlike object detection models, where adversarial training greatly reduces output curvature, adversarially robust segmentation models show only slightly reduced output curvature compared to their non-adversarially robust counterparts (Fig 3), and all *increase* curvature compared to the input. Investigating curvature change over model layers (Fig 8) reveals that this trend is consistent throughout all layers of the model. This result highlights the importance of task on straightness, and supports the idea that curvature reduction is not simply the result of adversarial training; only when temporal stability is task-beneficial does adversarial training promote straightness. We also evaluated the final layer for each of these models, finding similar results (Sec A.3).

Another DNN task for which output predictions are low level is next-frame prediction for video sequences. As in image segmentation models, good predictions are expected to vary dynamically frame-to-frame, and representational straightness at and near the output layer is *not* a favorable quality in response to videos containing motion. Rather, for a frame-prediction network, a favorable quality would be equal curvature at the input and output. This is because for a well-performing model, the output prediction over many frames is the same as the input video; two identical videos have identical curvature. To evaluate this, we test the layer-evolution of curvature for three variants of PredNet (Lotter et al., 2016), a network inspired by predictive coding, which is trained to predict the next frame in a video sequence (Sec A.7, Fig 28 bottom).

Indeed, for all pretrained variants of PredNet tested, the output layer maintains a very similar curvature value to that of its input frame, with final output curvature closer than all other models to that of the input (Fig 3). For all pretrained variants, the model curvatures increase in the representation of the first model block, then re-straighten the representation throughout the rest of the network before returning to the original pixel curvature. We also investigate a variant of PredNet fine-tuned to predict multiple frames in the future (PredNet_Extrap_KITTI), which we find only slightly reduces model curvature, with a similar trajectory over layers. Since PredNet is trained on videos, we can directly evaluate the impact of out-of-distribution compared to in-distribution data on straightness (Fig 28). PredNet models straighten more on in-distribution data, but the effect is small and does not change the overall trend seen in the videos from (Hénaff et al., 2019)). We also test adding a curvature constraint as a loss during training (see Section A.7) and find that while the model can straighten, it does not greatly impact the final model output (Figure 29).

## 7 CURVATURE IN BIOLOGICALLY INSPIRED MODELS

We investigated straightness for a variety of both parametric (learned) and non-parametric biologically-inspired models. Given straightness is thought to increase over progressively deeper layers of visual processing, we align these networks along the visual processing areas they are most closely matched to (Fig 9). Some models match to visual areas explicitly in their architecture like the non-parametric Henaffbio (Hénaff et al., 2021), which is a two-stage model based on center-surround filters follows by oriented Gabor filters, and PerpheralNet (previously referred to as Brucenet) (Brown et al., 2021), which is based on summary statistics of a steerable pyramid. This explicit match to biological vision also holds for the early layers VOneNetCornets (Dapello et al., 2020), which combines a V1 filter front-end and Cornets as a backbone (Cornets being a shallow neural network that has layers matched to regions of the human brain (Kubilius et al., 2019)). For the adversarially trained Visual Transformer network CrossViTRotAdv (Berrios & Deza, 2022), these layers are those best matched by BrainScore for V4 and IT layers (Schrimpf et al., 2020).

For all biologically-inspired models except for VOneNetCornets, curvature progressively decreases through deeper network layers. For VOneNetCornets, curvature decreases up until the V1 layer, where a noise term capturing neural stochasticity is added; curvature then strongly increases, far above the pixel-curvature baseline. To determine if this increase in curvature was due to the added noise, we tested the same model with the noise term set to zero. This reduced the downstream curvature after the V1 layer, but did not eliminate the curvature increase following the V1 layer.

The increase in curvature for VOneNet at later layers suggest that making the front-end of a deep network more like biologically-inspired models does not suffice to straighten representations downstream in a deep network. This is interesting because VOneNet is reported to be more adversarially robust to white-box attacks than a standard CNN. This suggests that adversarial *training*, not the property of adversarial robustness itself, leads to straightened representations. Our finding supports neural population geometry results showing that adversarial trained models have different signa-

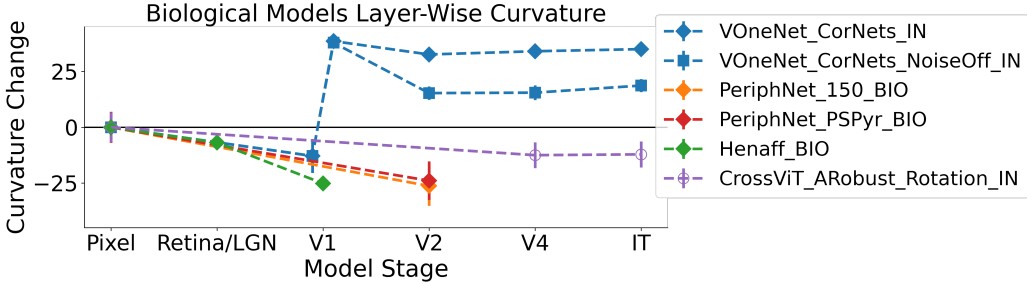

Figure 9: **Biologically-inspired network models result in straightened representations compared to input pixels.** Non-parametric multi-stage filter models PeriphNet and Henaffbio progressively straighten. The VOneNetCornets model straightens the representation in its V1-block up until its noise layer, where curvature increases dramatically and does not recover in downstream convolutional blocks. Adversarially-trained CrossViT model shows progressive straightening.

tures than VOneNet and models trained with neural stochasticity (Dapello et al., 2021). However, it is puzzling that adversarial training, which is biologically implausible, would lead to straighter representations than biologically inspired mechanisms. More constraints may be needed to achieve straight representations with biologically plausible methods.

## 8 DISCUSSION

We have shown that one can use change in model representational curvature as a simple and computationally cheap metric for evaluating certain types of robustness, as well as representational stability in both image and video models and across a variety of tasks. We have shown that training with strong white-box adversarial attacks reduces output curvature, and that models that reduce output curvature have better adversarial accuracy. However, incorporating simple shift-invariance and robustness to image corruption is not associated with straightness. Moreover, models that reduce the curvature of their input give more stable predictions over time. Furthermore, the relationship between curvature reduction and adversarial robustness depends on task. We find that adversarially robust object recognition models have straightened representations, but semantic segmentation and video frame prediction models do not. Moreover, in evaluating both CNN and transformer-based architectures, we have shown that the relationship between curvature reduction and adversarial robustness depends on task and training procedure, not deep neural network architecture alone.

In evaluating curvature for biologically inspired models, we show that biologically inspired mechanisms reduce curvature in a model's representation, even more than adversarial training. However, the simple addition of non-parametric biologically inspired filters at the input of a convolutional model is insufficient to maintain output curvature. These results identify representational straightness as a common thread between biologically inspired and adversarially robust models, highlighting the benefits and limitations of these techniques in creating temporally-stable representations.

With the growing need to understand the relationship between the robust properties of human vision and computer vision systems, straightness serves as an effective and efficient tool to evaluate and compare learned visual representations. With curvature reduction as a known property of the human perceptual representation, we find that our straightness results match intuitions such as adversarial robustness being important to achieving more human-like representations, and offers new insights such as the failure of self-supervised learning approaches at creating temporally stable, human-aligned representations. Curvature can also be computed layer-wise, allowing us to more deeply understand how model architectures support or impede the favorable properties of biologically-inspired representations. While many biologically-inspired networks have useful properties such as robustness, curvature reveals a more nuanced relationship between such networks and this robustness. Furthermore, curvature evaluates temporal stability while still being applicable to static image models. This allows for an expansion of current research aligning computer vision with human vision that has historically focused on static vision. We believe an essential next step in this research is to incorporate more temporal aspects of vision, and perceptual straightening is an important stepping-stone to get there.

## 9 REPRODUCIBILITY

To reproduce our results, links to all the models analyzed can be found in the supplemental material in the Network Comparison Spreadsheet. Sources for the stimuli are in Sec A.2.2 and in the code base linked for the HenaffBio model in Network Comparison Spreadsheet. Straightness can be determined by calculating curvature using Secs 3 and A.1. The models, stimuli, and curvature function together should be sufficient to reproduce our results.

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

# A   APPENDIX

## A.1   CURVATURE VS COSINE SIMILARITY

$$\text{cosine similarity (vectors)} := \cos(\theta) = \frac{A \cdot B}{\|A\|\|B\|} \tag{3}$$

$$\text{cosine similarity (video frames)} := \cos(\theta) = \frac{x_t \cdot x_{t+1}}{\|x_t\|\|x_{t+1}\|} \tag{4}$$

$$\text{curvature} := c_t = \arccos(\hat{v}_t \cdot \hat{v}_{t+1}) \tag{5}$$

$$\text{cosine curvature} := \hat{v}_t \cdot \hat{v}_{t+1} = \frac{v_t \cdot v_{t+1}}{\|v_t\|\|v_{t+1}\|} = \cos(c_t) \tag{6}$$

## A.2   MODELS

### A.2.1   MODEL SOURCES

All deep neural networks we analyzed were pretrained. Exact links to code and weights can be found in the supplemental material in the Network Comparison Spreadsheet. The standard ImageNet-trained ResNet (He et al., 2016) model was downloaded from PyTorch's model zoo (Paszke et al., 2019). Adversarially robust ResNet models were all downloaded from (Engstrom et al., 2019a). The adversarially robust ResNets were trained using projected gradient descent. All ViT (Dosovitskiy et al., 2020) and standard trained CrossViT daggar (Chen et al., 2021) models were downloaded from the timm library (Wightman, 2019). All CrossViT daggar adversarially robust and rotationally invariant models were downloaded from the repository of (Berrios & Deza, 2022). The adversarially robust CrossViTs were trained with fast gradient sign method as stated in (Berrios & Deza, 2022). DINO models were downloaded from the DINO repository (Caron et al., 2021), while PredNet models were downloaded from (Lotter et al., 2016).

### A.2.2   MODEL ANALYSIS PROCEDURE

We showed each model the same 12 natural videos that were used in the psychophysics experiments of (Hénaff et al., 2019). The videos were taken from the Chicago Motion Database (at University of Chicago, 2022), the film 'Dogville', Lions Gate Entertainment (2003), and LIVE Video Quality Database (Seshadrinathan et al., 2010b;a). The videos were grayscale, consisting of 11 frames each of $512 \times 512$ pixels, capturing natural motion such as rippling ocean water or a person walking through a crowded street. We resized the video frames to be $224 \times 224$ for all deep networks and $256 \times 256$ for bio-models that use steerable pyramids. One limitation of this work is that we did not evaluate models on larger video datasets, but we wanted to use psychophysically validated stimuli for our analyses. For each model, we recorded its activations at intermediate and final layers for each video. We then found the global curvature for each stage of the model using equation 2 where we used the flattened model activations as the input $x_t$ to the curvature procedure.

We compared the global curvature at each layer of the model to the curvature of the video in pixel space. Models that straighten are defined as models that have a lower global curvature at deeper layers. When comparing the curvature of different model layers, we chose not to reduce the dimensionality of each layer activation to be the same across stages. Although principle components analysis (PCA) was sometimes used in (Hénaff et al., 2019) when expressing curvature, they did not use it in their analysis of deep networks. Furthermore, while an architecture's inherent dimensionality is likely relevant to a representation's curvature, we preferred not to introduce any additional transformations that would influence the measured curvature. We also found that performing PCA did not greatly affect trends like adversarial training leading to more straight representations See Figure 10 for examples of curvature for different numbers of principle components on two ResNet50 models.

### A.2.3   COMPUTE

Our methods do not require large compute. All individual model analyses can be run on CPU. We used a single GPU to speed up getting the features activations at each layer to the order of minutes per model.

## A.2.4 EFFECT OF PCA ON MODEL CURVATURE

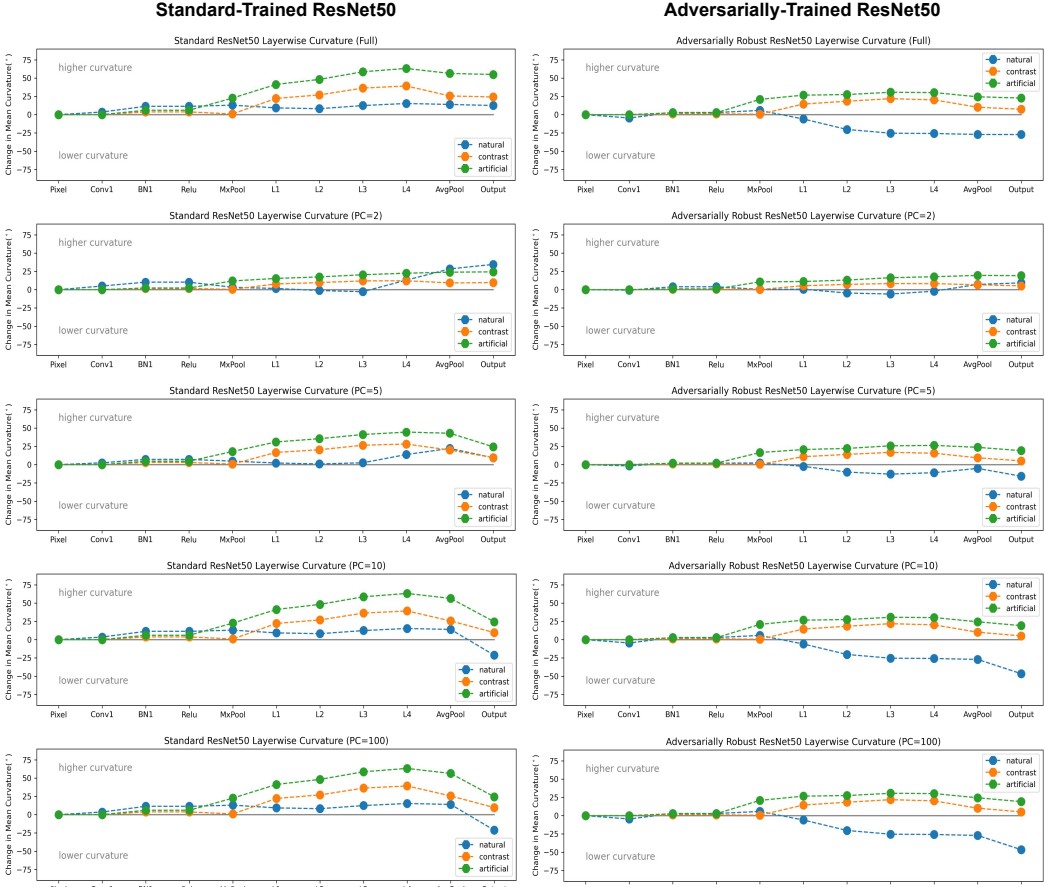

Figure 10: **Change in curvature for different numbers of principle components**. Each subplot shows layer-wise change in curvature. **Left** is a standard ResNet50 trained on ImageNet (Deng et al., 2009) classification. The natural line is the curvature measures on the natural videos from (Hénaff et al., 2019) that we used for all model analyses in the paper. The contrast and artificial sequences were additional video sequences psychophysically tested by (Hénaff et al., 2019) where humans tend to have little change in curvature for contrast but large increase in curvature for artificial. **Right** is an adversarially robust (projected gradient descent with $l_2$ norm and $\epsilon = 3$) ResNet50 trained on ImageNet. All rows below the top row show curvature when taking the first 2, 5, 10, and 100 principle components (PCs) of each representation layer activations. For PCs less than the number of frames in each video (11), we performed PCA on the pixel-level video in addition to the model activations when computing the change in curvature. For all PCs, the standard ResNet50 has a higher curvature for natural video sequences than the adversarially robust.

## A.3 FINAL STAGE CURVATURES

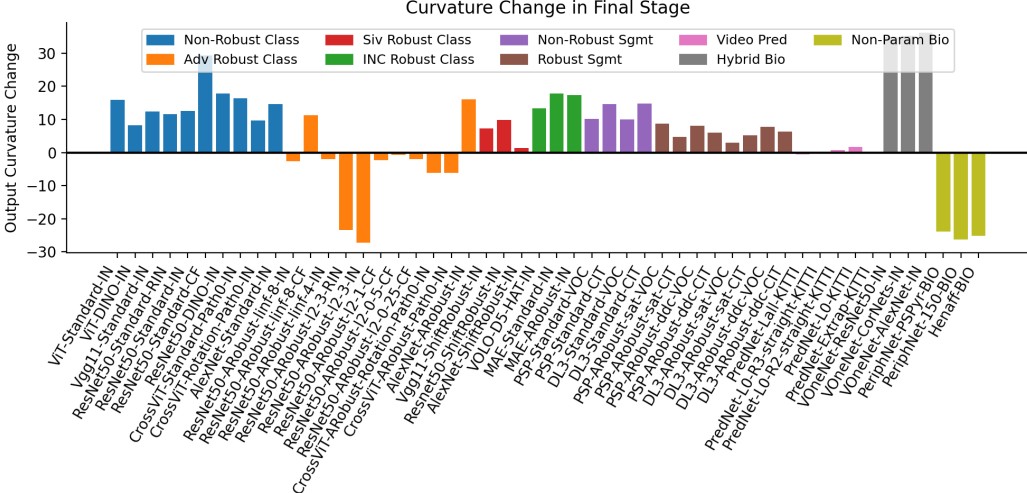

Figure 11: **Final stage curvature for a variety of network architecture and training schemes.** Model families show very similar curvature to that of their output representation layers 3.

The final stage of a network for image classification models is the last fully connected layer. We see similar trends in curvature across model families as in Fig 3. We chose to report the the curvature on the layer just before classification because this is the layer that would be most commonly used as the representational backbone for other tasks.

## A.4 MATCHING MODELS CURVATURE COMPARISON PLOTS

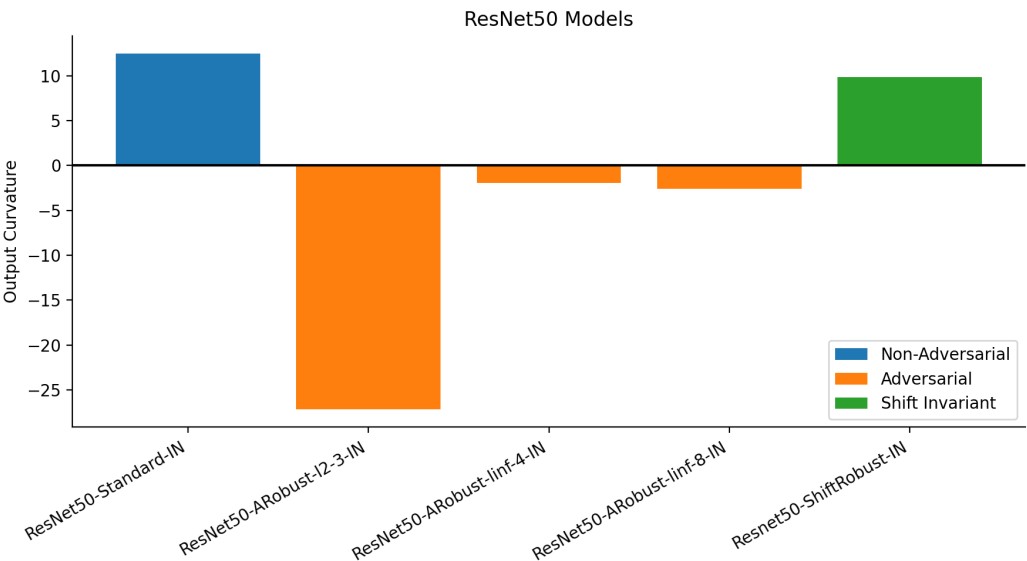

Figure 12: ResNet50 models trained on ImageNet

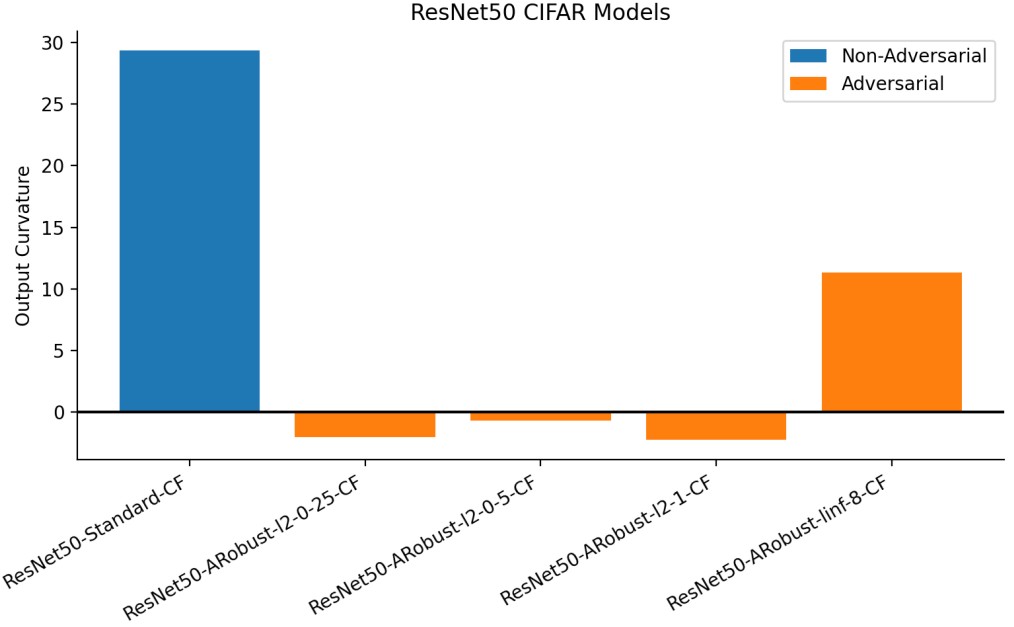

Figure 13: ResNet50 models trained on CIFAR10

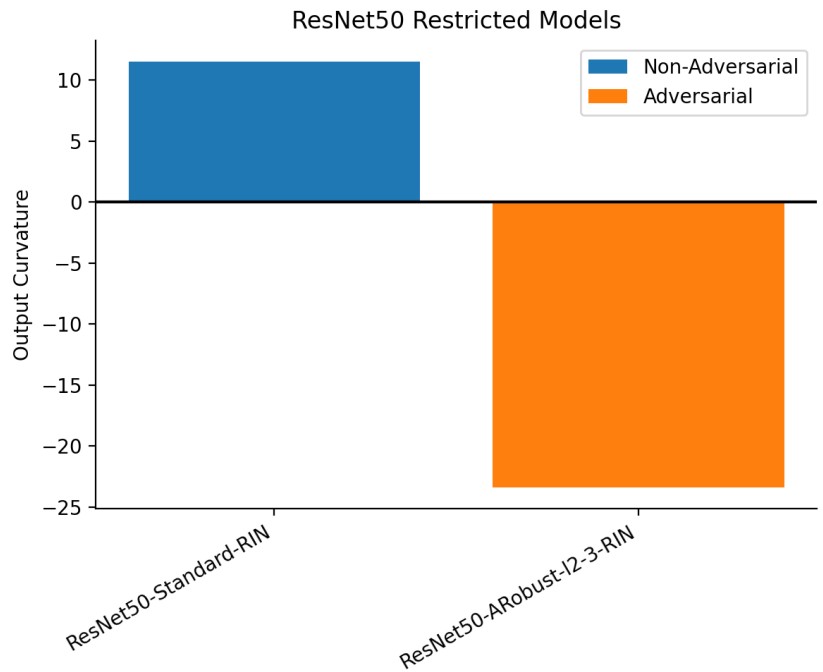

Figure 14: ResNet50 models trained on Restricted ImageNet (a subset of ImageNet)

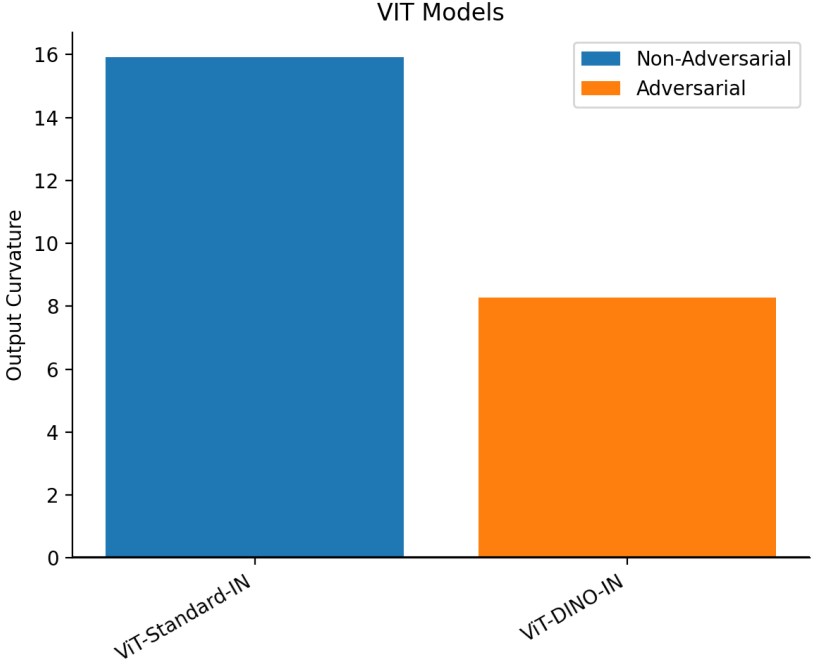

Figure 15: ViT models trained on ImageNet

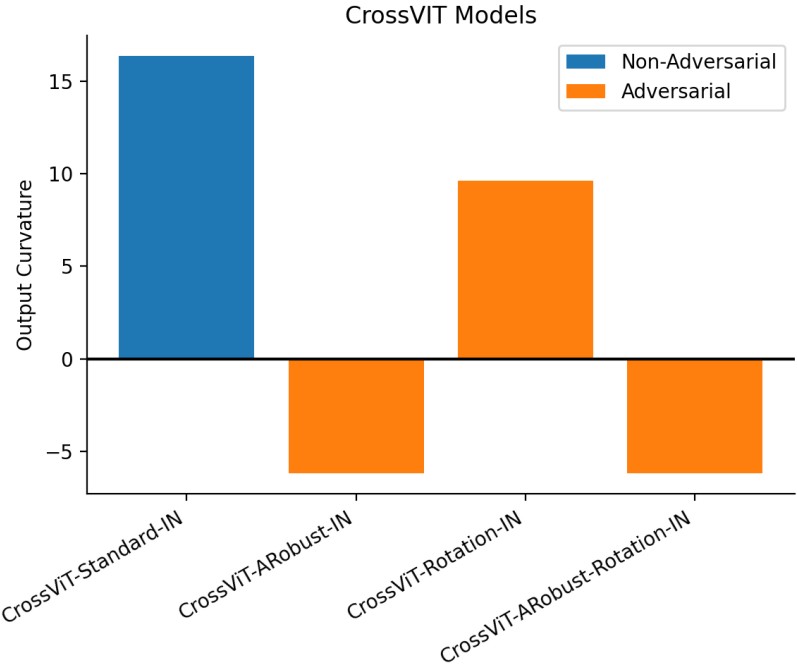

Figure 16: CrossViT models trained on ImageNet

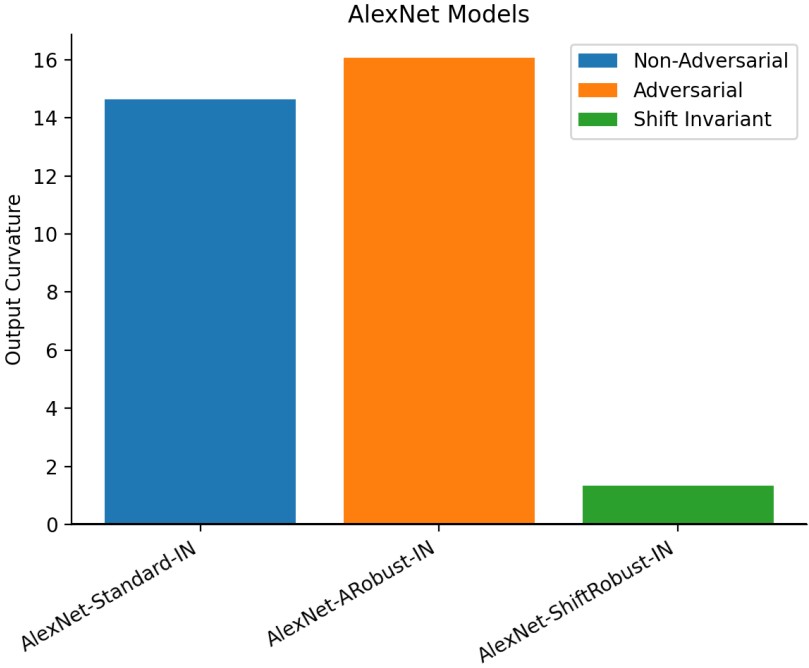

Figure 17: AlexNet models trained on ImageNet

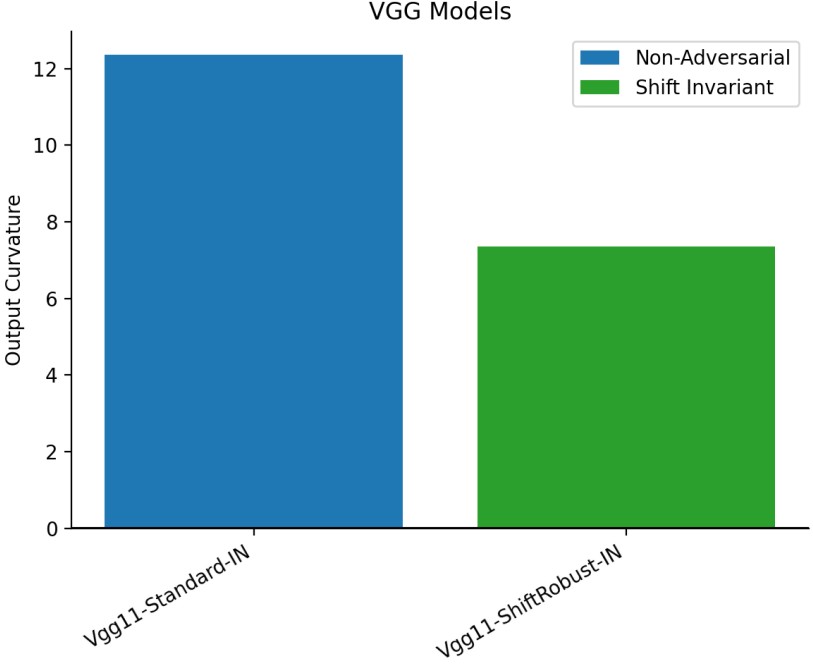

Figure 18: VGG models trained on ImageNet

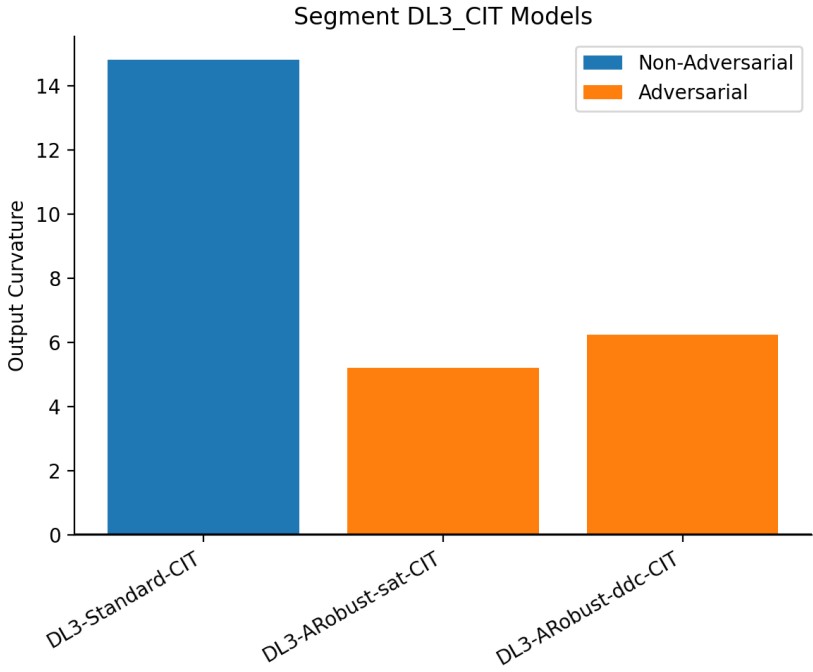

Figure 19: DL3 segmentation Models (Cordts et al., 2016)

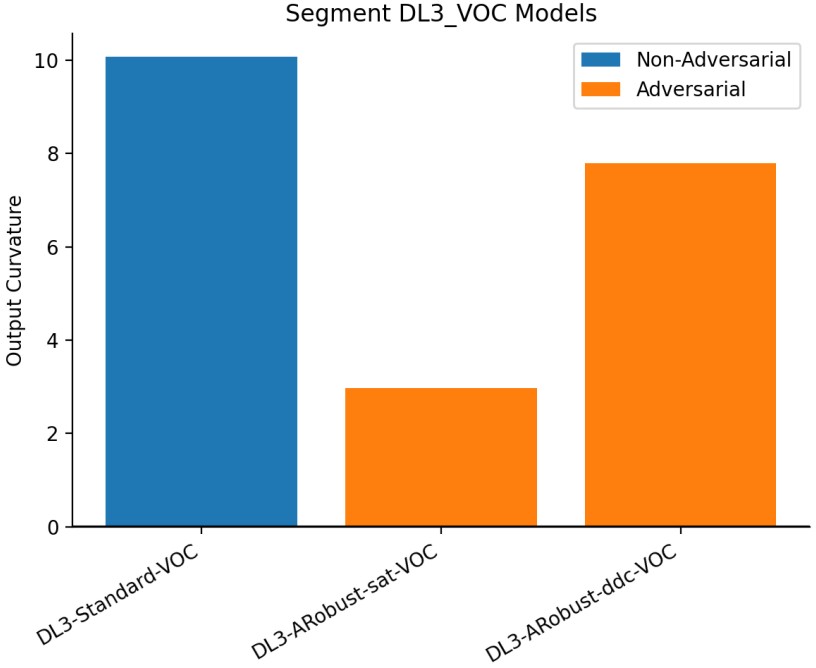

Figure 20: DL3 segmentation Models trained on VOC (Everingham et al., 2010)

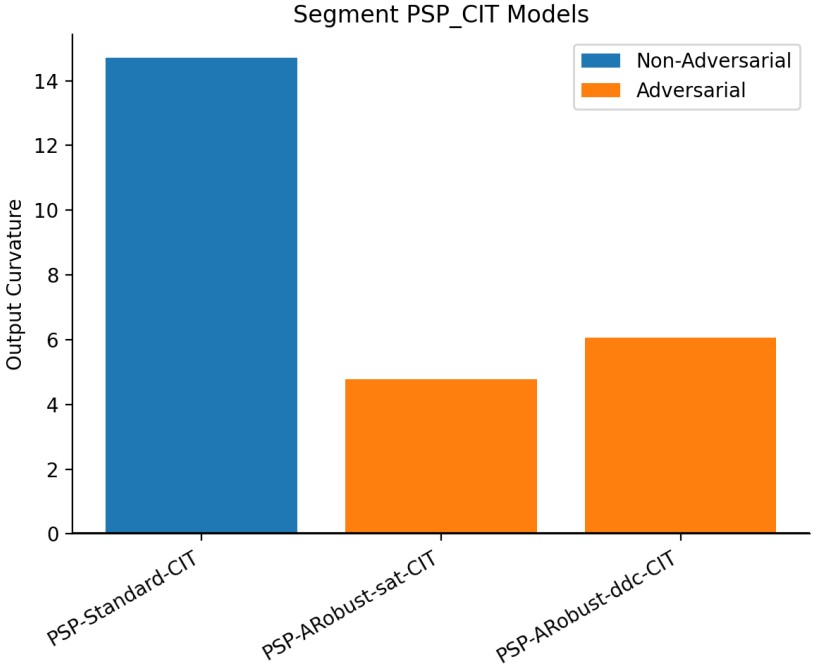

Figure 21: PSP segmentation models trained on CIT (Cordts et al., 2016)

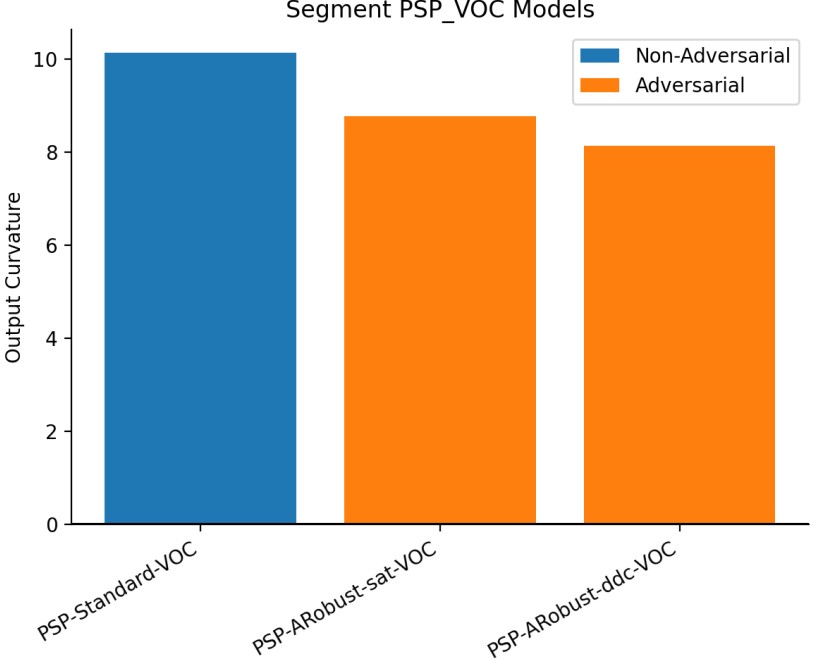

Figure 22: PSP segmentation models trained on VOC (Everingham et al., 2010)

## A.5 ADVERSARIAL ACCURACY AND CURVATURE

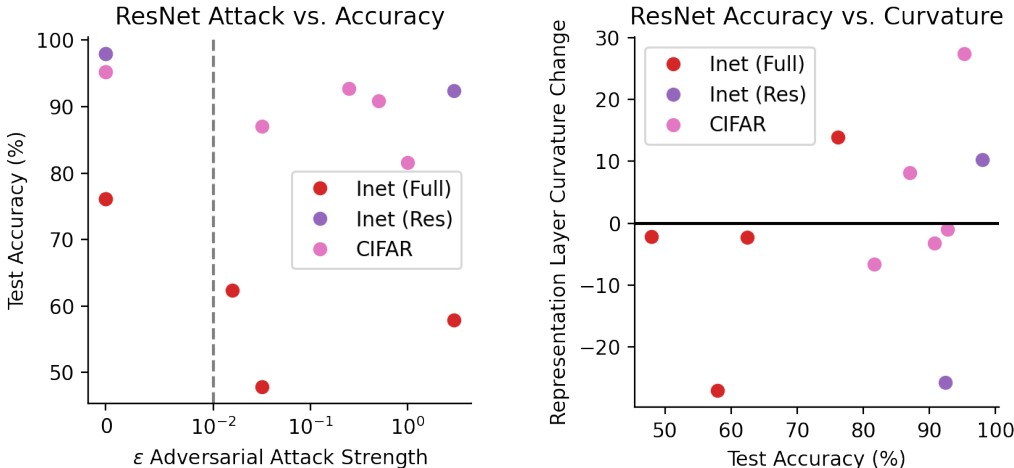

Figure 23: **Left**: While adversarial attacks with greater $\epsilon$ strength impart many desirable robustness properties on a network, adversarial training does not improve test accuracy, often decreasing test accuracy on the within-domain test set for a given model. (Data plotted on symlog scale.) **Right**: While stronger adversarial attacks decrease curvature, improved test accuracy for a model is not predictive of output curvature reduction. Rather, within a given model training/test set, increased test accuracy predicts a smaller curvature reduction in the output layer.

## A.6 CURVATURE IN NON-ADVERSARIALLY ROBUST NETWORKS

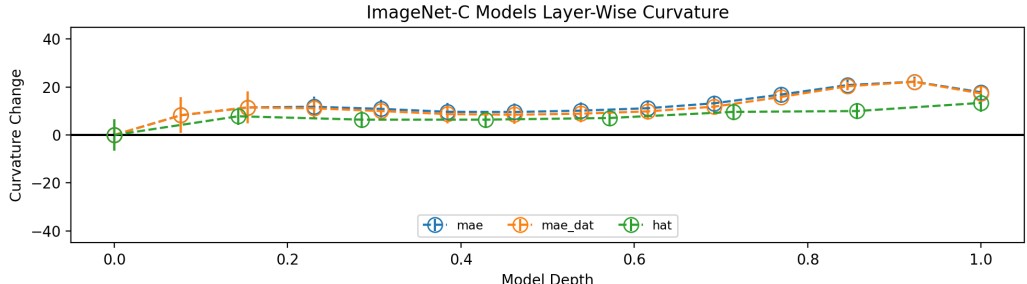

Figure 24: Curvature plots for three of the top five performing models for ImageNet-C (corrupted) dataset benchmark. MAE is a masked autoencoder transformer network (He et al., 2022), mae_dat is the mae model with additional discreet adversarial training (Mao et al., 2022) (note this is a modified version of traditional adversarial training for visual models), and hat is a visual transformer optimized to process high-frequency image components Bai et al. (2022). Although these models are robust to image corruptions, they do not reduce curvature. The mae_dat and hat models also represent the top two performing models on Stylized ImageNet, indicating that robustness to shape/texture transforms like those in Stylized ImageNet (Geirhos et al., 2018) is also not indicative of curvature reduction.

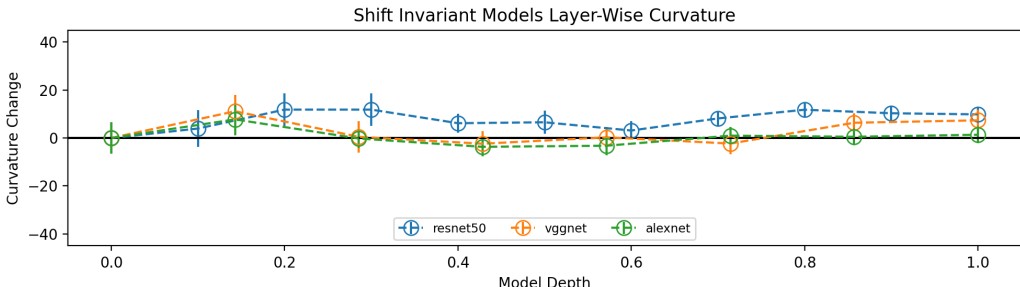

Figure 25: Curvature plots for three model architectures modified to be shift-invariant using the anti-aliasing method from (Zhang, 2019). These models are robust to location shifting, but do not show significant curvature reduction from input, as seen in adversarially-trained robust models

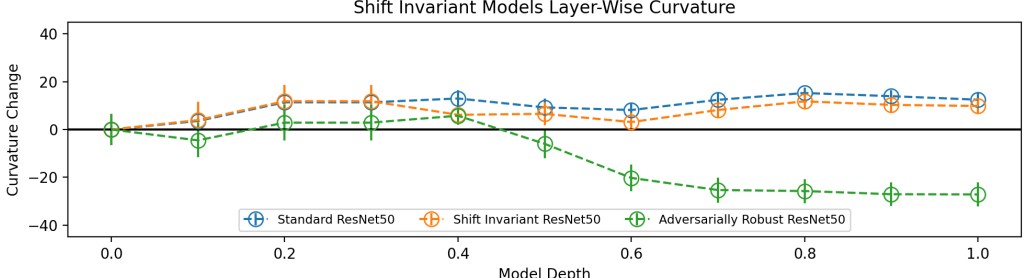

Figure 26: Curvature plots for ResNet50 variants trained: without robust training (Standard ResNet50), with shift-invariance robustness as in (Zhang, 2019) (Shift Invariant ResNet50), and with adversarial training (Adversarially Robust ResNet50).**Shift invariant robustness only slightly reduces curvature as compared to the standard model, and does not reduce output curvature from the input data, while adversarial robustness strongly reduces curvature reducing it below that of the input data.**

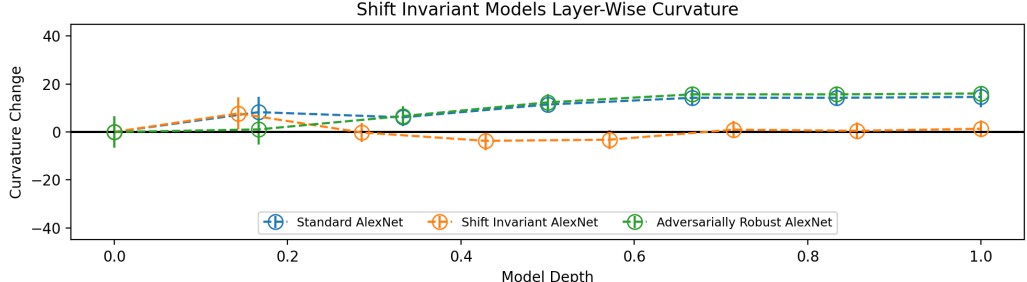

Figure 27: Curvature plots for AlexNet variants trained: without robust training (Standard AlexNet), with shift-invariance robustness as in (Zhang, 2019) (Shift Invariant AlexNet), and with adversarial training (Adversarially Robust AlexNet (Liu et al., 2021)). The small size of AlexNet limits its ability to benefit from adversarial attacks (Liu et al., 2021). This may explain why, unlike other adversarially trained models tested, curvature is not reduced significantly from the input. Shift-invariance in this model however, does appear to reduce output curvature as compared to the standard model.

## A.7 PredNet Straightness for In and Out of Distribution Data

We measured straightness in PredNet, next-frame video prediction model. In Figure 28, urvature is shown in filled circles for the following standard PredNet models: standard (PredNet_L0_KITTI), 5-frame future prediction (PredNet_Extrap_KITTI), and all-layer-loss (PredNet_Lall_KITTI). Shown in open circles are two PredNet models trained with straightness terms in the loss function (Pred-Net_L0_R2_straight_KITTI and PredNet_L0_R3_straight_KITTI). All models were trained on the KITTI dataset (Geiger et al., 2012).

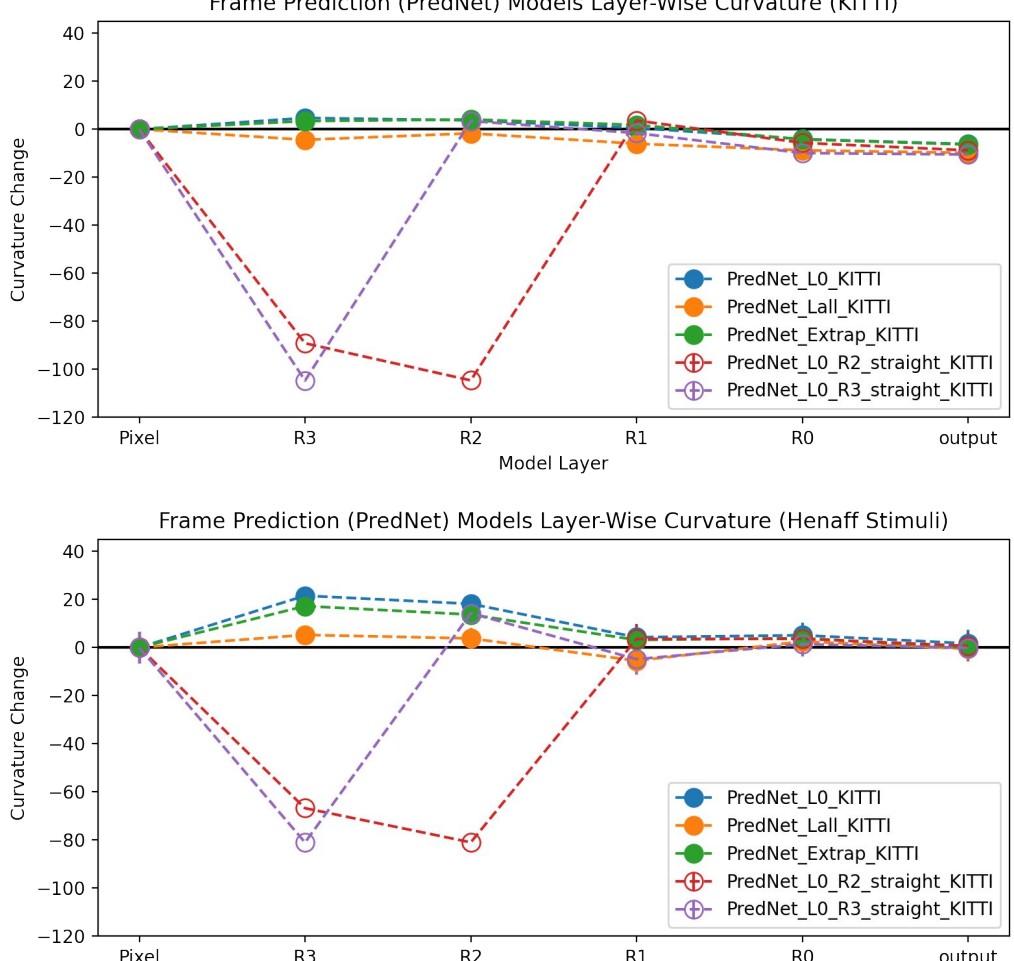

Figure 28: **PredNet change in curvature across model layers measured in-distribution (top: KITTI dataset) and out-of-distribution (bottom: (Hénaff et al., 2019) stimuli)**. Generally, all PredNet models have lower curvature values for KITTI, the in-distribution dataset, than the (Hénaff et al., 2019) stimuli, but the curvature follows similar trends across dataset. All PredNet models are trained on KITTI. The blue, orange, and green lines are the baseline PredNet models developed in (Lotter et al., 2016). The baseline models have a lower change in curvature on the KITTI dataset than the (Hénaff et al., 2019) stimuli. However, the change in curvature is generally close to 0 for both datasets – indicating little straightening. The red and purple lines show PredNet models that were trained to straighten layers 2 and 3 respectively on the KITTI dataset. Those models show a larger drop in curvature on the KITTI dataset than (Hénaff et al., 2019) stimuli at layers 2 and 3, but the models overall straighten in only two layers for both datasets.

The two models in Figure 28 that straighten (red and purple lines) are trained with a curvature constraint. Our goal is to tested the utility of straightness as a training signal by adding a curvature constraint to the first (R3) and second (R2) layers of PredNet during training to create "straight" PredNet models. We evaluate how the constraint affects the model's internal representation and the final frame prediction. For both models, curvature is greatly reduced in R2 and R3 compared to the input curvature (Fig 28 bottom). Because of PredNet's top-down and bottom-up feedback structure, the constraint on layer R3 does not affect any other layers of the model. Although we find that it is possible to train PredNet to straighten and still get next-frame predictions that has lower mean square error (MSE) than copying the previous frame, we are not able to achieve equivalent or better prediction MSE with the straight models compared to the original on KITTI (Geiger et al., 2012) (original L0 PredNet: 0.00687 MSE, straight R2 PredNet: 0.00976 MSE, compared to previous frame: 0.02122 MSE). See Fig 29 for example predictions. While these straightened PredNet variants do not achieve state of the art, we argue that these models may display other advantages such as robustness and similarity to human representations

### A.8    PREDNET PREDICTIONS

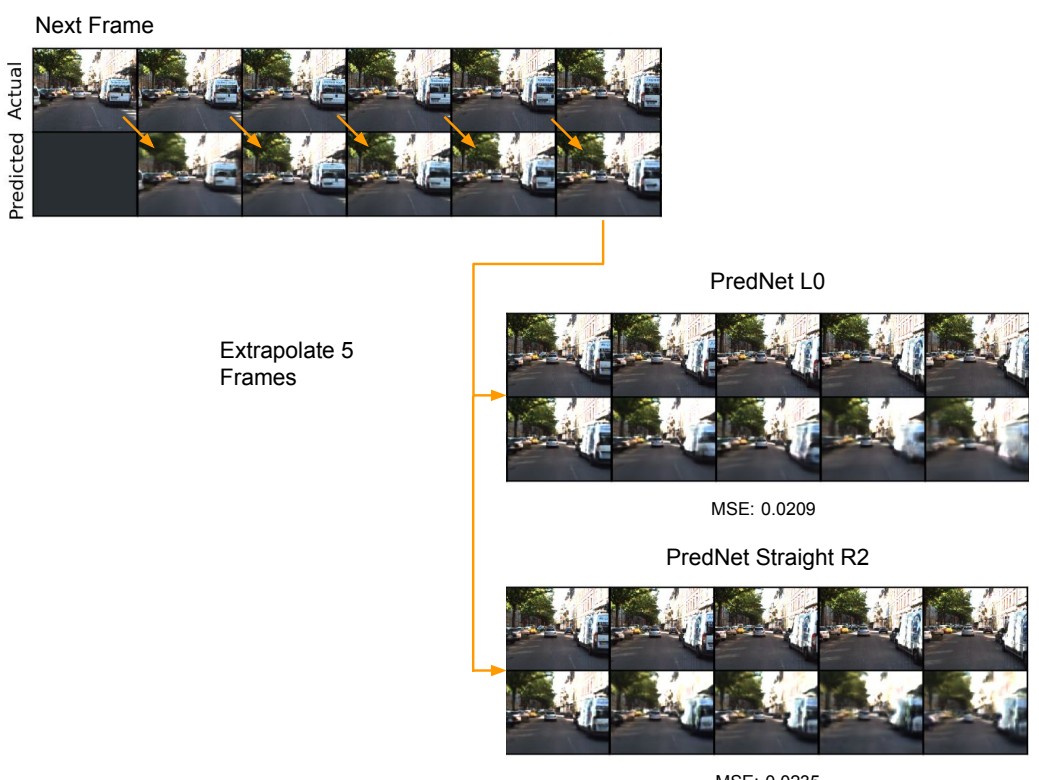

Figure 29: **Example Predictions for Original and Straighten PredNet**. Both models were trained on the KITTI dataset (Geiger et al., 2012). The straightened PredNet was trained with penalty on its R2 layer if there was high curvature.

ACKNOWLEDGMENTS

We thank Ila Fiete and Nick Watters for helpful discussions and feedback. A.H. is supported by funding from Toyota Research Institute. V.D. is supported by the MIT CSAIL METEOR Fellowship. This work was funded in part by NSF/BMBF IIS-1607486 to Ruth Rosenholtz and Christoph Zetzsche and by USAFRL and the United States Air Force Artificial Intelligence Accelerator under Cooperative Agreement Number FA8750-19- 2-1000 to W. Freeman. The authors acknowledge the MIT SuperCloud and Lincoln Laboratory Supercomputing Center for providing (HPC, database, consultation) resources that have contributed to the research results reported within this paper.

NEGATIVE SOCIETAL IMPACTS

We believe there are few negative societal impacts of this paper. Our work was exploratory and did not introduce any new models. However, we note that the development of machine vision systems, which can operate at or above the level of performance of humans at certain tasks, may lead to negative societal impacts such as the loss of jobs and industries as human workers are replaced.

