# OpenReview forum: "Exploring perceptual straightness in learned visual representations"
_ICLR.cc/2023/Conference — ICLR 2023 poster_

### Official Review · Reviewer_eEJj · 2022-10-24

**Confidence:** 4
**Correctness:** 3
**Technical Novelty And Significance:** 2
**Empirical Novelty And Significance:** 2
**Recommendation:** 6

**Clarity, Quality, Novelty And Reproducibility:**

- The presented work is written with great clarity. Figures that explain the authors methods such as Figs 1, 2 were very useful to understand the paper's main theme. Some of the result figure legends however lacked clarity as highlighted in my reviews.
- This work is a novel extension of prior work analysing representational straightness in the context of adversarial training. I don't find any big reproducibility issues per se, however, the code and weights for segmentation models isn't available in the Network Comparison Spreadsheet in the supplemental data.

**Strength And Weaknesses:**

Strengths:
1) Clarity of writing and figures. The paper is written with great clarity. Figures such as Fig 1 and Fig 2 are elegant and clear to convey what the authors are studying in this work.
2) Diverse evaluation. The authors have evaluated the representational straightness in a wide range of diverse deep neural network architectures from the robust and non-robust classes for image recognition and segmentation. They also evaluate multiple biologically plausible models that are of great interest to the visual neuroscience and vision science communities.
3) Fig 5, $\epsilon$ vs curvature change. I found this result to be quite interesting that increasing adversarial attack strength also increases straightness. It would be nice to see how much each of these attack strengths contributes to adversarial robustness as well.

Weaknesses:
1) Significance of the contribution to representation learning. The authors have mentioned in several places how straighter representations bear more advantages than test accuracy, such as robustness, alignment to human perception, and representational stability. However, I don't find any experiment/analysis that quantifies these advantages or even demonstrates them. Is representational straightness predictive of improved robustness to adversarial attacks and OOD shift? While it is obvious that adversarial training improves robustness to adversarial attacks, I would like to see the relationship between adversarial robustness and representational straightness. Similarly one could also correlate straightness with model generalization to OOD shift on benchmarks such as ImageNet-c. These evaluations would add more information about how significant the straightness changes are and how they relate to the representation quality.

2) Non-matching models in robust and non-robust class. Since one of the main takeaways of the paper is that adversarial training improves straightness, I expect to see clearly how the same architecture (and same rnd seed) -- when fixing other factors of variability such as dataset, training set size, training hyperparameters -- varies in their representation straightness. There are some matching models in the robust and non-robust classes in Fig 3. If the above conditions of varying adversarial training in an isolated manner was what was performed for this analysis, I would suggest the authors please make this clear in writing.

3) Hard to understand model specifications from legends. I believe in some figures (e.g Fig 4, 5, 6) it is difficult to know the combination of architecture + training dataset. I would suggest the authors to use a common template to define models e.g. Architecture_NumberOfLayers_Dataset_Task and it would make reading the plots much easier.

4) Discussion section. Currently the discussion section presents a summary of the results in the paper (how curvature changes as a function of adversarial training, optimization task etc.) but does not discuss in detail why these observations are relevant to designing future deep learning models or to human visual representations. This section could be greatly improved to highlight why, in the authors' perspective, the results are of importance to representation learning and human vision.

**Summary Of The Paper:**

The authors present an analysis of deep learning models using the change in curvature across frames of videos from pixel space to the representation space. The authors find that training deep networks on classification with adversarial images leads to straighter (like human visual representations, Henaff et al 2019) representations. They also analyze biologically plausible networks to show that there is some (in the early layers) increase in straightness but not consistent across the complete hierarchy. I find the presented analyses to be interesting, but I don't think the authors have clearly conveyed how their contributions impact either machine learning or vision science.

**Summary Of The Review:**

I think the paper presents an interesting set of results linking representational straightness to adversarial training. However, without any evidence on a) how straightness better predicts human visual behavior/neural data or b) how straightness leads to more adversarially or OOD robust deep learning models, I am presented with a partial picture of the significance of the contributions. At this stage, I find the paper to be borderline and would be comfortable recommending acceptance if the authors make the suggested changes mentioned in my review above. Mainly, addressing the following points (quoting from my review) and how the results turn out to be will influence a change in my score:

"Is representational straightness predictive of improved robustness to adversarial attacks and OOD shift? While it is obvious that adversarial training improves robustness to adversarial attacks, I would like to see the relationship between adversarial robustness and representational straightness. Similarly one could also correlate straightness with model generalization to OOD shift on benchmarks such as ImageNet-c. These evaluations would add more information about how significant the straightness changes are and how they relate to the representation quality."

---

> ### Author Response · Authors · 2022-11-15
> **Initial author response and clarification (part 1 for 2)**
>
> We thank reviewer eEJj for the thoughtful review of our paper, and for helpful suggestions that will improve the paper’s impact.
>
> We value the reviewer’s request for *“experiment/analysis that quantifies these advantages (of straightness) or even demonstrates them. Is representational straightness predictive of improved robustness to adversarial attacks and OOD shift? While it is obvious that adversarial training improves robustness to adversarial attacks, I would like to see the relationship between adversarial robustness and representational straightness.”* To address these questions, and further demonstrate the importance of curvature as a useful way to evaluate network representation, we have included an additional experiment (see Section 4.2) identifying a beneficial property of networks with lower curvature. We show that for a ResNet50 network evaluated on the VGGSound dataset, the reduced curvature property resulting from adversarial training also results in more stability for label predictions, even for out-of-distribution datasets. This is a strong advantage for a network.
>
> We appreciate this and another reviewer’s suggestion to *“correlate straightness with model generalization to OOD shift on benchmarks such as ImageNet-c. These evaluations would add more information about how significant the straightness changes are and how they relate to the representation quality.”* We are currently evaluating curvature for top models on ImageNet-C and anti-aliasing CNNs. We report preliminary results at the end of Section 4.1 and in Appendix section A5. We thank the reviewers for suggesting this addition which we expect to improve the paper significantly.
>
> We thank the reviewer for pointing out ambiguity regarding *“Non-matching models in robust and non-robust class. Since one of the main takeaways of the paper is that adversarial training improves straightness, I expect to see clearly how the same architecture (and same rnd seed) -- when fixing other factors of variability such as dataset, training set size, training hyperparameters -- varies in their representation straightness.”* In response, we have downloaded pretrained models from previous work, and do not have the resources to retrain all of them with exactly matched hyperparameters. However, this matching has been done by previous authors for some models such as the CIFAR trained ResNet50 models used by (Madry et al. 2019) and provided in the Robustness library, where all hyperparameters are controlled for, with only adversarial attack strength changing. In addition, most adversarially trained models are fine-tuned on the standard matched model they are derived from; we consider this the most fair comparison we can achieve without training all models from scratch. We will modify our manuscript to make it clear where additional factors such as dataset size and hyperparameters are controlled for in matching models, such that curvature in the variants can be fairly compared. Would the reviewer like to also see a plot like figure 3, but comparing matching models side by side? We again thank the reviewer for calling this to our attention.
>
> We appreciate the reviewer’s suggestions to *“use a common template to define models” that would “make reading the plots much easier”*. We will update these labels throughout the paper and make them more consistent in our revision.
>
> We also thank the review for pointing out that *“the code and weights for segmentation models isn't available in the Network Comparison Spreadsheet in the supplemental data.”* We have updated our manuscript to include links to the pretrained segmentation model weights, and will include links to weights for our new anti-aliasing models.
>
> Continued in following comment ...

---

> > ### Author Response · Authors · 2022-11-15
> > **Initial author response and clarification (part 2 of 2)**
> >
> > Like other reviewers, reviewer eEJj feels that while our paper is clear in findings, they *“don't think the authors have clearly conveyed how their contributions impact either machine learning or vision science.”* Likewise, we appreciate the reviewer’s suggestion to expand the discussion section as it does not yet *“discuss in detail why these observations are relevant to designing future deep learning models or to human visual representations. This section could be greatly improved to highlight why, in the authors' perspective, the results are of importance to representation learning and human vision.”* We are currently expanding the discussion section to make the significance of straightness more clear. We will emphasize that the curvature metric is a useful tool in evaluating neural networks for the following reasons:
> >
> > - There is a growing need to evaluate how human-like models are, and curvature can do this much more quickly and easily than other methods such as psychophysics experiments. Curvature reduction has both been shown to be a property of human perceptual representation (Henaff et.al), and we find a relationship between BrainScore and the curvature measure where models that do well on BrainScore such as the adversarially trained CrossVIT also perform well on the straightening measure.
> >
> > - We also believe that there should be greater awareness about perceptual straightening because it measures a temporal property of human vision and while still being applicable to static image models.  A lot of research in aligning computer vision with human vision computer vision has focused on static vision. We believe an essential next step in that research is to incorporate more temporal aspects of vision, and perceptual straightening could be an important stepping-stone to get there.
> >
> > - Curvature can be computed layer-wise, allowing us to more deeply understand the favorable properties of biologically-inspired representations. For example, VOnetNet has been shown to have useful properties such as robustness, but curvature reveal the nuanced relationship between biological inspired networks and this robustness (ie we show that curvature reduction is not maintained in convolutional layers downstream from the bio-inspired layers of VOneNet).
> >
> > - Curvature is extremely cheap to compute (order of minutes). We see that curvature can give a quick sense of how a computer vision model compares to humans without the need for additional psychophysics or physiology experiments
> >
> > - Curvature seems to be associated with adversarial robustness, a property useful to computer vision systems.

---

> > > ### Author Response · Authors · 2022-11-19
> > > **Updated paper submitted**
> > >
> > > We have uploaded a new version of the paper and the supplemental materials with the changes we promised.  We addressed all the points you made in your review which has greatly improve the paper as a whole. Thank you for your thoughtful feedback!

---

> > > ### Comment · Reviewer_eEJj · 2022-11-26
> > > **Acknowledgement of author responses**
> > >
> > > I thank the authors for careful consideration of the reviews and providing a thorough response. I find the newly added VGGsound analysis to quantify representational stability to be quite interesting but it must be noted that this analysis is still preliminary to assess stability.
> > >
> > > I thank the authors for also clarifying on my question about matching model hyperparameters, I do agree that re-training models by deliberately matching parameters will be resource-heavy. However like the authors rightly mention, making possibly biasing differences between compared models explicit in the paper would be really appreciated. To your question on whether I would like to see matching models compared side by side in Fig 3's style, YES! That would be a very fruitful analysis to perform and add to the paper.
> > >
> > > I agree with the important advantage of curvature evaluation that the authors raise. Typical measurements of similarity to human-like visual representations aren't cheap to compute, unlike representational curvature which indeed can be straightforward to compute and won't require as much resources. This, taken in combination with the observation that curvature correlates with relatively more compute heavy metrics like Brain Score as the authors mention in the response is quite encouraging.
> > >
> > > Overall, I raise my score to 6 and recommend acceptance. I think the results especially after the author response do carry value to the audience interested in human-like visual representation learning. However I'm not giving a strong accept because the analyses could be strengthened further (e.g. 1) more fleshed out stability study, 2) ImageNet-c evaluation to study the relationship between perceptual robustness and curvature, and 3) maybe also looking at the model uncertainty as a function of curvature (here is a great overview I found on measuring uncertainty, it looks like there are quite a few overlapping interests here with developing human-like visual representations)). Thank you again to the authors for the submission and for engaging in the response with meaningful improvisations to the paper.

---

> > > > ### Author Response · Authors · 2022-11-30
> > > > **Thanking Reviewer and Clarifications**
> > > >
> > > > Thank you for your time in reading our response and updates, and for your response. We are happy to have addressed many of your concerns.
> > > >
> > > > 1) We agree a more fleshed out stability study would be beneficial, and plan this for future work.
> > > > 2) Is the reviewer aware of the ImageNet-c experiment that we have included in the revision (Supplemental, A6), evaluating two top performing models on Imagenet-C for curvature reduction?  Or is the reviewer suggesting a more fleshed out experiment in this area?
> > > > 3) This is an interesting idea, to relate perceptual straightness to model uncertainty. We are unable to see the link for the overview on measuring uncertainty you mention. Could you post it again?
> > > >
> > > > Finally, regarding side by side models in the style of figure 3, we have added side-by-side comparisons of the models which share pre-trained models as a baseline (Supplemental, A4).
> > > >
> > > > Thank you again for your help in improving our paper.
> > > > Authors

---

### Official Review · Reviewer_zqwQ · 2022-10-25

**Confidence:** 2
**Correctness:** 3
**Technical Novelty And Significance:** 3
**Empirical Novelty And Significance:** 3
**Recommendation:** 6

**Clarity, Quality, Novelty And Reproducibility:**

The paper is easy to follow, and the empirical evaluation appears and novel to me.

**Strength And Weaknesses:**

Strength:
-	The proposed study appears novel to me and is well motivated.
-	 Authors perform an extensive evaluation on impressive number of models.

Weaknesses:
-	While the study is motivated by finding in biological system, it’s unclear why perceptual straightness is a desirable property from a machine learning perspective. It would be nice to identify a set of tasks where improved straightness has a benefit, beyond adversarial robustness.
-	What is the impact of the dataset distribution on which we compute the output curvature? The evaluation protocol only uses 12 videos to compute this metric. Would the finding remain stable if we were to use different videos?
-	Videos are most likely out-of-distribution with respect to the training dataset. Would the finding be similar if we were to reduce the distribution shift between training and evaluation?



**Summary Of The Paper:**

This paper investigates the perceptual straightness of a large set of modern vision models. Authors start from the observation that representation straightness is a known property of biological vision. They then define a metric to capture the average output curvature on a video and evaluate a wide range of models (resnet/vit trained for classification, segmentation models and biologically inspired model).

The main conclusions of the study are:
1)	Non-adversely trained models have the highest output curvature
2)	Adversarial training can reduce the model output curvature, when trained with classification tasks but not for all the tasks, i.e. it is not case for segmentation.
3)	Biologically inspired mechanism can also reduce the model output curvatures.


**Summary Of The Review:**

The paper proposes an interesting empirical study. The significance of the paper could be improved by better demonstrating the advantage of representation with a higher level of straightness and showing that the conclusions are robust to the choice of the video dataset used in evaluation.


=== After reading rebuttal.

Thank you for your responses. The rebuttal addressed most of my original concerns, I updated my score accordingly.

---

> ### Author Response · Authors · 2022-11-15
> **Initial author response**
>
> We thank reviewer zqwQ for their thoughtful review of our paper, and for helpful suggestions that will improve the paper’s impact.
>
> We appreciate the reviewer’s insights regarding in- and out-of distribution data: *“Videos are most likely out-of-distribution with respect to the training dataset. Would the finding be similar if we were to reduce the distribution shift between training and evaluation?”*  Because curvature is a metric that must be tested with videos, we cannot test in-distribution for most of the tested networks, as they are trained on image datasets. However, for the 5 PredNet variants tested, which are trained on Kitti, we have now evaluated curvature change for both the Henaff videos (out of distribution), as well as for the original Kitti dataset videos (in distribution) (see Appendix Section A6 and paragraph 4 of Section 6). We find that for PredNet variants, curvature results are fairly consistent between in and out of distribution test videos. Overall, the curvature is further reduced for the in-distribution Kitti dataset, as opposed to the out of distribution Henaff dataset that we tested the networks on. Notably, for the PretNet variants fine-tuned with a straightness term in the loss function for intermediate layers, straightness is only slightly reduced for the out-of-distribution Henaff dataset.
>
>
> Reviewer zqwQ notes that *"It would be nice to identify a set of tasks where improved straightness has a benefit, beyond adversarial robustness”*, and that “*The evaluation protocol only uses 12 videos to compute this metric. What is the impact of the dataset distribution on which we compute the output curvature? Would the finding remain stable if we were to use different videos?”* We appreciate their insight in these points. To address these concerns, we have included a new experiment where we evaluate straightness and prediction stability of two ResNet50 models on the VGGSound dataset to increase the variety of dataset used (see Section 4.2). We show that  the reduced curvature property resulting from adversarial training also results in more stability in correct label predictions, even for a larger out-of-distribution dataset.
>
> Like other reviewers, the reviewer zqwQ feels that while our paper is clear in findings, it is not clear *“why perceptual straightness is a desirable property from a machine learning perspective.”* We appreciate the reviewers calling this to our attention, and are revising the paper to make these points more clear. We emphasize that the curvature metric is a useful tool in evaluating neural networks for the following reasons:
>
> - There is a growing need to evaluate how human-like models are, and curvature can do this much more quickly and easily than other methods such as psychophysics experiments. Curvature reduction has both been shown to be a property of human perceptual representation (Henaff et.al), and we find a relationship between BrainScore and the curvature measure where models that do well on BrainScore such as the adversarially trained CrossVIT also perform well on the straightening measure.
>
> - We also believe that there should be greater awareness about perceptual straightening because it measures a temporal property of human vision and while still being applicable to static image models.  A lot of research in aligning computer vision with human vision computer vision has focused on static vision. We believe an essential next step in that research is to incorporate more temporal aspects of vision, and perceptual straightening could be an important stepping-stone to get there.
>
> - Curvature can be computed layer-wise, allowing us to more deeply understand the favorable properties of biologically-inspired representations. For example, VOnetNet has been shown to have useful properties such as robustness, but curvature reveal the nuanced relationship between biological inspired networks and this robustness (ie we show that curvature reduction is not maintained in convolutional layers downstream from the bio-inspired layers of VOneNet).
>
> - Curvature is extremely cheap to compute (order of minutes). We see that curvature can give a quick sense of how a computer vision model compares to humans without the need for additional psychophysics or physiology experiments
>
> - Curvature seems to be associated with adversarial robustness, a property useful to computer vision systems.

---

> > ### Author Response · Authors · 2022-11-19
> > **Updated paper submitted**
> >
> > We have uploaded a new version of the paper and the supplemental materials with the changes we promised.  We addressed all the points you made in your review which has greatly improve the paper as a whole. Thank you for your thoughtful feedback!

---

> ### Author Response · Authors · 2022-11-30
> **Thanking the Reviewer**
>
> Thank you for your time reading our response and paper updates, and for appreciating our revision. We are very happy to have resolved your concerns.
>
> Please feel free to share any further comments or concerns with us.
>
> Thank you again for your help in improving our paper,
> Authors

---

### Official Review · Reviewer_EBgB · 2022-10-26

**Confidence:** 4
**Correctness:** 3
**Technical Novelty And Significance:** 1
**Empirical Novelty And Significance:** 3
**Recommendation:** 6

**Clarity, Quality, Novelty And Reproducibility:**

Clarity:
The paper is clearly written.
Some citations are not in parentheses while they should be.
The text fontsize in the figures should be almost as large as the main text fontsize.
The figures in the supplementary are not referred to in the main text. As such they are useless. Other supplementary sections should also be referred to in the main text.


Quality:
My opinion is that it is great to have neural network architectures that are aligned with visual perception and neurosciences. Yet, the interest of the paper might be limited to an already acquired audience if the authors do not provide more explanations about why straightening is useful or interesting to have in NNs.
The performed work is of good quality. Yet, as it is a single focus paper I would expect to find a broader evaluation of robustness and not only adversarial training.

Novelty:
Evaluation of straightening in neural network representation is novel but this is the single focus of this paper.


Reproducibility:
Seems reproducible if the code is made available online.

**Strength And Weaknesses:**

Strength :
- straightening in neural network hasn't been previously evaluated,
- the work provides an overview of straightening in many architectures trained on many tasks,
- the work establishes a link between straightening and robustness,
- single focus paper with a clear message.

Weaknesses :
- the claims about robustness are weak because only adversarial training is used while there are other ways to enforce robustness,
- the authors failed to explain why "curvature is a useful way of evaluating neural networks representations".

**Summary Of The Paper:**

The authors propose to leverage a recent finding about human visual perception of movies to evaluate a variety of neural network architectures trained on different tasks. The visual feature is that humans tend to perceive movies with relatively less curvature from frame to frame compared to the actual curvature measured on pixels. We say that visual perception is straightening movies. The proposed work has three contributions :
(i) adversarial training often leads to straightening in CNN and Vision Transformers,
(ii) straightening depends on the training task (eg no straightening for segmentation),
(iii) biologically-inspired piece of architecture always leads to straightening when going deeper and deeper.

**Summary Of The Review:**

For now, I slightly tend to reject the paper. Yet, I will reconsider after reading authors responses and others discussions among reviewers.

---

> ### Author Response · Authors · 2022-11-15
> **Initial author response and clarification**
>
> We thank the reviewer for their thoughtful and thorough review of our paper, as well as the helpful suggestions that will broaden our paper’s impact.
>
> First, we wish to clarify a point with the reviewer regarding the stated contribution "*(iii) biologically-inspired piece of architecture always leads to straightening when going deeper and deeper.*" We in fact show the opposite, that bio-inspired models do NOT always lead to straightening, and that the straightening effect depends on the biological transform, and downstream layers.
>
> Secondly, we wish to address a stated weakness: “*the claims about robustness are weak because only adversarial training is used while there are other ways to enforce robustness*,” We greatly appreciate this and another reviewer recognizing additional types of robustness (other than adversarial) that we have not yet reported curvature results for. We are currently evaluating curvature for additional types of robust networks including top models on  ImageNet-C and anti-aliasing CNNs.  We report preliminary results at the end of Section 4.1 and in Appendix section A5.  Furthermore, we currently are revising the paper to be more specific about our claims or curvature’s relation to specific types of robustness, rather than robustness overall. We thank the reviewers for suggesting this addition which we expect to improve the paper significantly.
>
> Like other reviewers,reviewer EBgB feels that while our paper is clear in findings, we have “*failed to explain why ‘curvature is a useful way of evaluating neural networks representations’*", and that we should provide “*more explanations about why straightening is useful or interesting to have in NNs*”. We appreciate the reviewers calling this to our attention, and are revising the paper to make these points more clear. We plan to emphasize that the curvature measure is a useful tool in evaluating neural networks for the following reasons:
>
> - There is a growing need to evaluate how human-like models are, and curvature can do this much more quickly and easily than other methods such as psychophysics experiments. Curvature reduction has both been shown to be a property of human perceptual representation (Henaff et.al), and we find a relationship between BrainScore and the curvature measure where models that do well on BrainScore such as the adversarially trained CrossVIT also perform well on the straightening measure.
>
> - We also believe that there should be greater awareness about perceptual straightening because it measures a temporal property of human vision and while still being applicable to static image models.  A lot of research in aligning computer vision with human vision computer vision has focused on static vision. We believe an essential next step in that research is to incorporate more temporal aspects of vision, and perceptual straightening could be an important stepping-stone to get there.
>
> - Curvature can be computed layer-wise, allowing us to more deeply understand the favorable properties of biologically-inspired representations. For example, VOnetNet has been shown to have useful properties such as robustness, but curvature reveal the nuanced relationship between biological inspired networks and this robustness (ie we show that curvature reduction is not maintained in convolutional layers downstream from the bio-inspired layers of VOneNet).
>
> - Curvature is extremely cheap to compute (order of minutes). We see that curvature can give a quick sense of how a computer vision model compares to humans without the need for additional psychophysics or physiology experiments
>
> - Curvature seems to be associated with adversarial robustness, a property useful to computer vision systems.
>
> Finally, to further demonstrate the importance of curvature as a useful way to evaluate network representation, we have included an additional experiment (see Section 4.2) identifying a beneficial property of networks with lower curvature. We show that for a ResNet50 network evaluated on the VGGSound dataset, the reduced curvature property resulting from adversarial training also results in more stability for label predictions, even for out-of-distribution datasets.
>
> We also appreciate the reviewer’s attention to detail in pointing out the citation formatting, text font size in figures, and missing references to the supplemental in the main text. We are revising our paper to address these issues.
>
> For reproducibility, we will be releasing our code online.

---

> > ### Author Response · Authors · 2022-11-19
> > **Updated paper submitted**
> >
> > We have uploaded a new version of the paper and the supplemental materials with the changes we promised.  We addressed all the points you made in your review which has greatly improve the paper as a whole. Thank you for your thoughtful feedback!

---

> ### Author Response · Authors · 2022-11-29
> **Request for Reviewer's Response to Revisions**
>
> We again thank reviewer EBgB again for their careful initial review, which has helped us improve our paper significantly. With the time window closing on the discussion period, we hope to have further conversation with you. In particular, we hope to hear your response to the new experiments and updates which we believe address the weaknesses you noted in the previous version of the paper.
> In our revised paper, we evaluate curvature for additional types of robustness including top performing models on Imagenet-C which are robust to corruptions, as well as shift-invariant robust models (Supplemental Section, A6). We also clarify the type of robustness referred in different sections throughout the paper. An additional experiment shows curvature reduction's relationship to label stability (Figure 6), showing a measurable utility of the curvature measure. A second additional experiment shows curvature reduction associated with stronger AutoAttack accuracy (Figure 5, Right), demonstrating another utility of the curvature measure. We have also revised the introduction and conclusion to explain more clearly why curvature is useful in evaluating neural network representations. We also address the formatting issues noted by the reviewer (citations in parentheses, increased text font size in figures and references to supplementary material in the main body).
> We appreciate you reviewing our responses, and the updates to the paper.
> Thank you,
> Authors

---

### Decision · Program_Chairs · 2023-01-20

**Decision:**

Accept: poster

**Justification For Why Not Higher Score:**

As mentioned above, better benchmarks and more experiments on those benchmarks are needed to strengthen the impact of the paper.

**Justification For Why Not Lower Score:**

The paper proposed a valuable study, so the AC and the reviewers believe the community will benefit from such study.

**Metareview: Summary, Strengths And Weaknesses:**

The paper addresses the problem of "straightness" in perception models. Straightness in the context of videos refers to a smooth path between frames in the representation space. The paper has four main findings: (1) Adversarial training results in more straightened representations in vision architectures. (2) Increase in straightness by adversarial training is task dependent (e.g., not true for segmentation). (3) Straightening makes models more stable against OOD data. (4) Certain classes of biologically inspired models increase straightness in the lower layers of the network.

Strengths:
- The study of straightness is novel and well-motivated.
- The paper includes an extensive set of experiments considering a wide variety of models and tasks.
- The paper is well-written and conveys the message clearly.

Weaknesses:
- Evaluation benchmark only includes 12 videos, so it is not clear how generalizable these findings are.



**Note From Pc:**

if the above contains the word "oral" or "spotlight" please see: "oral" presentation means -> notable-top-5% and "spotlight" means -> notable-top-25%. As stated in our emails, we are disassociating presentation type from AC recommendations

**Summary Of Ac-Reviewer Meeting:**

The paper initially received three 5’s as the rating. The main criticisms were (1) The impact of the study on machine perception or vision science was not clear. (2) More evaluations on different benchmarks were needed to draw stronger conclusions. The authors addressed the concerns and ran additional experiments and included the preliminary results. The AC and the reviewers had a video call to discuss the paper (one of the reviewers did not participate despite the positive response for participation). Everybody agreed that the paper has merit and should be accepted, but more rigorous experiments are needed to strengthen some of the conclusions. All reviewers changed their rating to 6. The AC follows the recommendation of the reviewers and recommends acceptance.